



# Spectral proper orthogonal decomposition of active wake mixing dynamics in a stable atmospheric boundary layer

Gopal R. Yalla[1], Kenneth Brown[1], Lawrence Cheung[2], Dan Houck[1], Nathaniel deVelder[1], and Nicholas Hamilton[3]

[1]Sandia National Laboratories, Albuquerque, NM, USA
[2]Sandia National Laboratories, Livermore, CA, USA
[3]National Renewable Energy Laboratory, Golden, CO, USA

**Correspondence:** Gopal R. Yalla (gyalla@sandia.gov)

**Abstract.** Recent advancements in the use of active wake mixing (AWM) to reduce wake effects on downstream turbines open new avenues for increasing power generation in wind farms. However, a better understanding of the fluid dynamics underlying AWM is still needed to make wake mixing a reliable strategy for wind farm flow control. In this work, a spectral proper orthogonal decomposition (SPOD) is used to analyze the dynamics of coherent flow structures that are induced in the wake

through blade pitch actuation. The data are generated using the Exawind software suite to perform a large eddy simulation of an AWAKEN 2.8 MW turbine operating in a stable atmospheric boundary layer. SPOD tracks the modal behavior of flow structures from their generation in the turbine induction field, through their growth in the near wake region, and to their subsequent evolution and energy transfers in the far wake. SPOD is shown to be a useful tool in the context of AWM because it translates the wavenumber and frequency inputs to the turbine controller to structures in the wake. A decomposition of the

radial shear stress flux in the wake is also developed using SPOD to measure the contribution of coherent flow structures to mean flow turbulent entrainment and wake recovery. The effectiveness of AWM is connected to its ability to excite inherent structures in the wake of the turbine that arise using baseline controls. The effects of AWM on blade loading are also analyzed by connecting the axial force along the blade to the SPOD analysis of the turbine induction field. Lastly, the performance of different AWM strategies is demonstrated in a two-turbine array.

## 1 Introduction

Interactions between wind turbines and surrounding wakes often result in reduced power generation and increased structural

loads for downstream turbines in a wind farm (Nygaard, 2014; El-Asha et al., 2017). Power losses are particularly problematic in stable atmospheric boundary layers (ABLs), such as those found offshore, because of the increased persistence of wind



turbine wakes in these conditions. Several wind farm flow control (WFFC) methods have been proposed in the last decade to reduce the negative impacts of wake momentum loss in wind farms, including turbine derating, wake steering, and wake mixing (Meyers et al., 2022). These methods rely on static or dynamic adjustments to the operation of upstream turbines,
away from their optimal set-point, for the benefit of downstream turbines. This paper focuses on the technique of active wake mixing (AWM), which is a particularly promising approach for wake mitigation as it provides a mechanism for introducing new momentum into a wind farm. Specifically, AWM aims to excite coherent structures in the wake through dynamic oscillations in the turbine's control parameters to enhance the entrainment of higher velocity flow into the wake, resulting in faster wake recovery.

As with any WFFC strategy, AWM introduces power and load trade-offs to the wind farm optimization problem. Recent experimental and numerical results indicate that AWM can improve the power production of turbine arrays anywhere from 1 to 30% (Frederik et al., 2020c, b; Yılmaz and Meyers, 2018; Taschner et al., 2023; Frederik et al., 2020a), at the cost of increasing loads on upstream turbines due to pitch actuation (Frederik and van Wingerden, 2022). These results vary significantly with turbine layout, turbine model, and ABL condition. Additionally, the design space for AWM is considerably larger than that of
other WFFC strategies. Common implementations rely on at least four relevant design parameters to control the blade pitch fluctuations, which also result in significant performance differences (Cheung et al., 2024b). Given the complex interactions between wind conditions, wind farm layout, and turbine control parameters, we therefore cannot expect to rely solely on parametric studies to optimize AWM. Instead, a deeper understanding of the physical mechanisms behind the power and load trade-offs of AWM is necessary to effectively navigate the design space.

AWM introduces periodic oscillations in the turbine's control parameters, such as the blade pitch, to generate large-scale coherent structures in the wake. Several approaches have been developed in the recent literature to analyze the dynamics induced by these large-scale structures and their connection to the performance of different AWM strategies. Brown et al. (2025) analyzed the mean flow kinetic energy budget through a control volume analysis around the wake of an actuated turbine. This analysis was used to explain the power increases for a two turbine array reported by Frederik et al. (2025). It was shown
that AWM enhances the turbulent and, in some cases, the mean-flow entrainment of momentum into the wake, and that veer has a large impact on the relative performance among AWM strategies. Korb et al. (2023) noted the importance of turbulent entrainment as well, but also showed that helical structures in particular can spread and deflect the wake deficit, leading to faster wake recovery. Munters and Meyers (2018) used an adjoint framework to determine an optimal perturbation to impart on the wake, which resembled an axisymmetric flow structure that enhanced existing vortical structures in the non-actuated
wake. Cheung et al. (2024b) introduced a normal mode representation of the coherent wake structures and used linear stability analysis to quantify the growth characteristics of the initial flow disturbances in the wake.

Despite these advancements, a complete understanding of the performance differences between AWM strategies has not been established, including their impact on wake dynamics and relationship to ABL characteristics, which limits the applicability of wake mixing technologies in practice. Fortunately, large-scale coherent structures are a well studied phenomena in the broader
context of turbulent flows (Hussain and Reynolds, 1970; Robinson et al., 1991; Ho and Huerre, 1984; Crow and Champagne, 1971; Fuchs et al., 1979), and several techniques have been developed to extract, quantify, and model coherent flow features



(Rowley and Dawson, 2017; Taira et al., 2017). The spectral proper orthogonal decomposition (SPOD), in particular, has proven to be a useful representation of space-time coherence in statistically stationary flows (Towne et al., 2018). Since its introduction by Lumley (1967), SPOD has been applied to a number of turbulent flows, including turbulent boundary layers

(Tutkun and George, 2017), jets (Citriniti and George, 2000; Gudmundsson and Colonius, 2011), and wakes (Tutkun et al., 2008; Araya et al., 2017). In the context of AWM, Cheung et al. (2024b) demonstrated in a canonical flow that SPOD could be used to track the modal growth of instabilities in the wake.

In this paper, the applicability of SPOD to AWM is strengthened further. The coherent structures induced by AWM are analyzed from the turbine induction field to the far wake region using high fidelity large eddy simulation (LES) data of a wind

turbine operating in stable ABL conditions. SPOD is used to quantify the energetic structures in the wake for each AWM strategy, as well as their interactions with other large-scale wake structures. Additionally, the SPOD analysis is connected to conventional wake mixing metrics through a modal decomposition of turbulent entrainment statistics. This formulation provides insights into the wake recovery mechanisms of different AWM strategies and their relative performance, which is subsequently demonstrated in a two-turbine array. The flow structures that arise in the wake of a baseline-operated turbine are

also analyzed through SPOD and connected to the performance of different AWM strategies.

The remainder of this paper is organized as follows: The methodology for the paper is discussed in Section 2 including the LES specifications, ABL precursor generation, turbine model, implementation of AWM in the turbine controller, and SPOD formulation. The results from the SPOD analysis are then discussed in Section 3.1 for several AWM strategies, which are connected to conventional wake mixing metrics in Section 3.2 and turbine performance metrics in Section 3.3. Finally,

conclusions are provided in Section 4.

## 2 Methodology

### 2.1 LES formulation

Large eddy simulations of a single wind turbine are performed using the ExaWind Nalu-Wind solver. Nalu-Wind employs an unstructured second-order control-volume finite-element discretization in space and a second-order backwards-differentiation

scheme in time to solve the implicitly-filtered incompressible Navier-Stokes equations. Additional details on the Nalu-Wind solver and its application to wind turbine simulations is given by Sprague et al. (2020) and Domino (2015). A one-equation model for the evolution of the subgrid turbulent kinetic energy (TKE) is used to represent the subgrid-scale effects on the resolved flow (Davidson, 1997). The simulation domain ranges from $0 \text{ m} \le x \le 3000 \text{ m}$, $0 \text{ m} \le y \le 750 \text{ m}$, and $0 \text{ m} \le z \le$ 960 m in the streamwise, lateral and vertical directions, respectively. A three-dimensional visualization of the flow on a subset

of the domain is shown in Figure 1. Three distinct levels of isotropic resolution are used to discretize the domain, which range from 5 m near the domain boundaries, to 2.5 m around the wake, and 1.25 m near the turbine, leading to 64 million total computational elements.

The streamwise and lateral boundary conditions are defined by an inflow condition extracted from an initial precursor simulation. The target conditions for the precursor simulation are derived from stable ABL measurements taken in 2021 as part





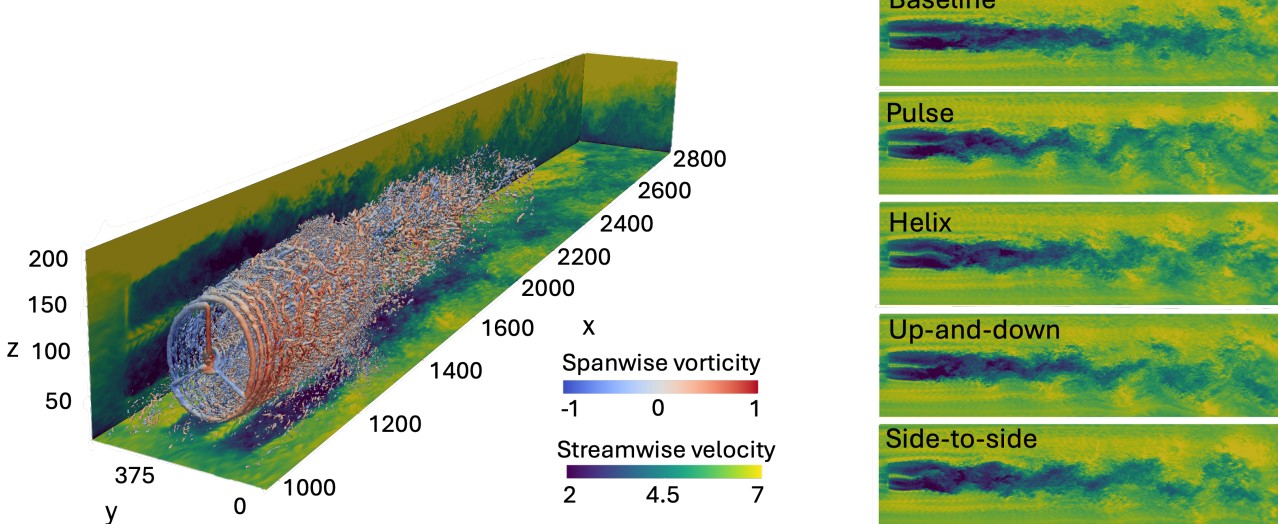

**Figure 1.** (Left) 3D visualization of the flow for the pulse case with $A = 1.25°$ pitching amplitude. Isosurfaces of the Q-criterion ($Q = 0.05$) are shown on a subset of the domain, colored by spanwise vorticity. Mid-planes of the wake are projected to the domain boundaries and show streamwise velocity. (Right) Contours of streamwise velocity for each AWM strategy on the hub height plane.

of the American Wake Experiment (AWAKEN) campaign (Moriarty et al., 2024). Specifically, the atmosphere was sampled at the Department of Energy's Atmospheric Radiation Measurement (ARM) Southern Great Plains (SGP) C1 site (see Figure 2) at heights of 10 m to 2000 m in increments of 26 m and reported at 10 minute intervals (Gaustad and Xie; Koontz et al., b, a; Shippert et al.; Jensen et al.; Keeler et al.; Kyrouac et al.) This data is then filtered to include wind speeds between 6 and 6.7 m/s, turbulence intensities up to 7%, wind directions ranging from $100°$ to $260°$, and a veer greater than $20°$ (see Figure 2). These

criteria are representative of typical stable ABL conditions, which are particularly suited to wake mitigation strategies such as AWM. Data from 230 minutes that meet these criteria are used to establish target statistics for the precursor simulation. The surface roughness height (0.0015 m) and cooling rate ($4.16 \times 10^{-5}$ K/s) parameters are calibrated in the precursor simulation of the ABL to ensure good agreement between the simulated and measured flow statistics (see Figure 3), although the simulated ABL exhibits noticeably less veer than the measurement data. The decrease in veer is due to the numerical forcing scheme used

to drive the ABL towards the target statistics (primarily TI and shear), which limits the amount of simulated veer that can be achieved while maintaining a realistic ABL velocity profile. This calibration results in inflow data with an average hub-height wind speed of $U_{\mathrm{inf}} = 6.4$ m/s, a turbulence intensity (TI) of 3.5%, a shear exponent of 0.17, and $9°$ of veer over the height of the rotor disk. The turbine is simulated within the ABL with a time step of $\Delta t = 0.02$ for 1,100s after reaching a steady state. Terrain effects are not included in either the precursor or the wind turbine simulations, and, instead, a flat lower surface

is implemented using the atmospheric rough wall model from Moeng (1984). A 100m wide inversion layer is applied to the temperature profile at $z = 700$ m to reduce perturbations in the flow above this height, and a temperature gradient of 0.00075





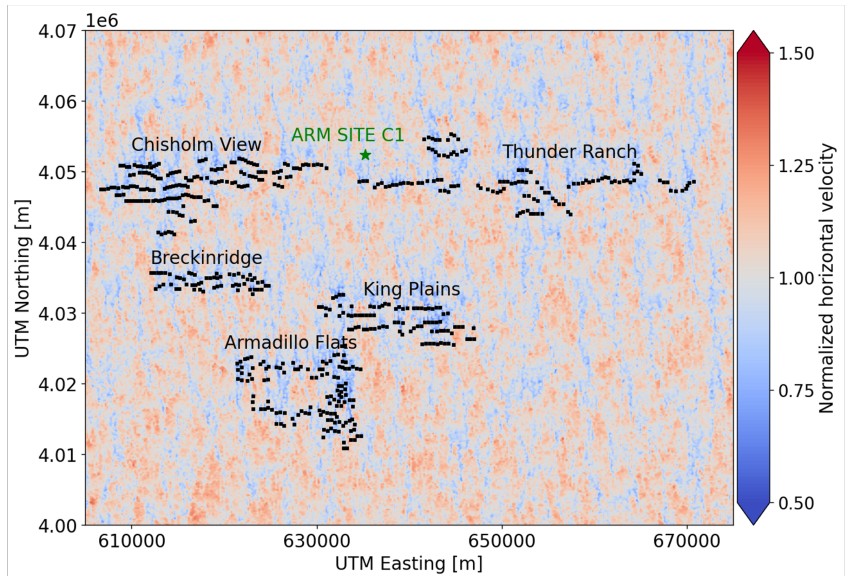

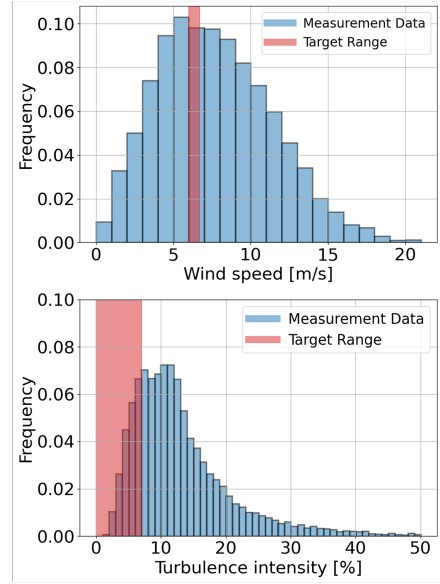

**Figure 2.** (Left) Site configuration for the five wind farms involved in the AWAKEN measurement campaign. The normalized velocity contours correspond to an unstable atmospheric boundary sampled at the hub-height plane (see Cheung et al. (2024a) and Moriarty et al. (2024) for more details). (Right) Histograms of the wind speeds at 91m and turbulence intensities at 60m sampled at the ARM SGP site C1 location over 230 minutes.

K/m is maintained at the upper boundary. A potential flow solution is used set the upper boundary condition for the velocity field.

The turbine specifications are provided by the NREL 2.8 MW turbine model (Quon, E. W., 2024), which provides an open-source representation of a turbine that is similar to those located at the AWAKEN site (see Figure 4). The hub of the turbine is located in the computational domain at $(x_{hub}, y_{hub}, z_{hub}) = (1050, 375, 88.5)$ m, within the $\Delta x = 1.25$ m resolution region. The wind speed in the precursor ABL simulation at the hub height location corresponds to region 2 for this turbine and also falls within the relatively constant thrust coefficient region (see Figure 4). To represent the dynamic response of the turbine, Nalu-Wind is coupled to the National Renewable Energy Lab's OpenFAST software suite (National Renewable Energy Laboratory, 2024b). An actuator line model (ALM) with an isotropic Gaussian spreading function is used to represent the turbine aerodynamic forces computed in OpenFAST as body forces in the LES (Sorensen and Shen, 2002). The filtered lifting line correction is applied with $\varepsilon/\Delta x = 2$ and $\epsilon^{opt}/c = 0.25$, where $c$ is the chord length, to improve the ALM's representation of the force distribution along the blade (Martínez-Tossas and Meneveau, 2019; Martínez-Tossas et al., 2024).

To implement different AWM control strategies on the turbine, a dynamic blade pitch, $\Theta$, is specified on top of the baseline pitch set point, $\Theta_0$. Specifically, the normal-mode representation of the blade pitch signal developed by Cheung et al. (2024b) is implemented in NREL's reference open-source controller (ROSCO v2.8.0; National Renewable Energy Laboratory (2024a)),





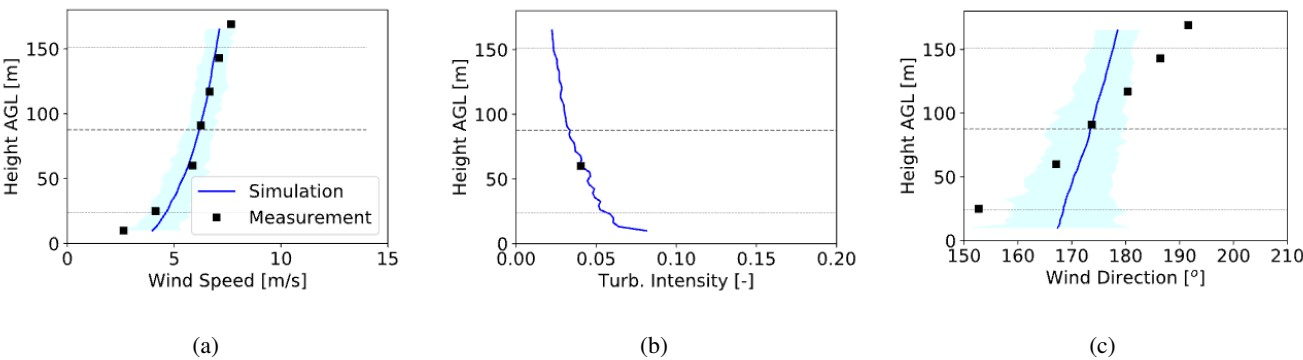

**Figure 3.** Horizontally averaged profiles of the simulated and measured (a) wind speed, (b) turbulence intensity, and (c) wind direction.

and coupled to OpenFAST in the LES. The dynamic blade pitch is specified as

$$\Theta(t) = \Theta_0(t) + A \sum_{\{\kappa_\theta\}} \cos(\omega_e t - \kappa_\theta \psi(t) + \phi_{\text{clock}}), \tag{1}$$

where $A$ is the pitching amplitude, $\omega_e$ is the excitation frequency, $\psi$ is the azimuth position of the blade, $\phi_{\text{clock}}$ is the clocking

angle, and $\kappa_\theta$ is an azimuthal wavenumber. The summation in equation 1 is over a discrete set of azimuthal wavenumbers. Of particular importance to this study are the $\kappa_\theta$ and $\omega_e$ parameters, which control the structure and frequency of the flow structures that are imparted on the wake, respectively. The excitation frequency can be specified through a Strouhal number based on the inflow velocity, $U_{inf}$, and turbine diameter, $D$, as $\omega_e = 2\pi St U_{inf}/D$. A Strouhal number of $St = 0.3$ is used for all control strategies considered here, as this has been shown to align with natural unsteady processes in the wake and result in

good performance for different AWM cases (Frederik et al., 2020c; Munters and Meyers, 2018; Cheung et al., 2024b). A single Strouhal period based on the inflow velocity and turbine diameter corresponds to 66.15s, which means that flow structures are generated over much longer periods than a typical rotor period (see Figure 5). The 1,100s of simulation time corresponds to over 16 complete Strouhal cycles at $St = 0.3$.

Four AWM strategies are considered in addition to a baseline case, including a pulse ($\kappa_\theta = 0$), helix, ($\kappa_\theta = -1$), and side-

to-side ($\kappa_\theta = \pm 1$) actuation, all with $\phi_{\text{clock}} = 90°$, as well as an up-and-down ($\kappa_\theta = \pm 1$) actuation with $\phi_{\text{clock}} = 0°$. Here, positive and negative azimuthal wavenumbers denote flow structures that rotate in the same and opposite direction of the turbine, respectively (i.e., clockwise and counter-clockwise when looking downstream). The counter-clockwise helix method is used here, which has been found to consistently outperform its clockwise counterpart (Coquelet et al., 2024). The pulse forcing is axisymmetric and achieved through collective pitching of the three turbine blades, while the other AWM strategies

rely on individual pitch control to create a non-uniform thrust force around the rotor disk that varies with the clocking angle and wavenumber. A pitching amplitude of $A = 1.25°$ is primarily used in this study, although other values of $A$ are also discussed. It is important to note that for the up-and-down and side-to-side cases, a pitching amplitude of $A = 1.25°$ is applied to both the $\pm 1$ modes, resulting in twice the total pitch amplitude as the pulse and helix cases. A summary of the AWM parameters is



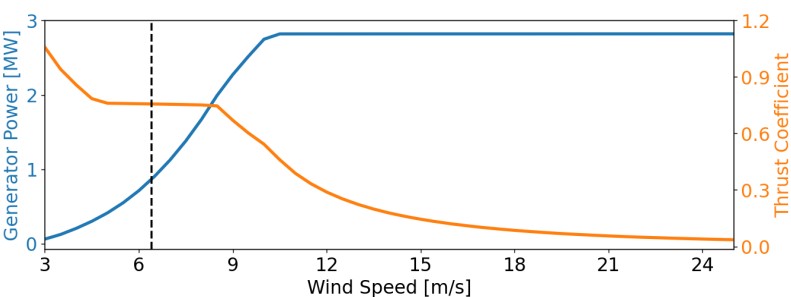

| NREL 2.8MW | |
| --- | --- |
| Hub-height (m) | 88.5 |
| Rotor diameter (m) | 127 |
| Rated RPM | 12 |
| Design Ct | 0.75 |
| Rated TSR | 8 |

**Figure 4.** Specifications of the NREL 2.8MW reference turbine model (Quon, E. W., 2024) including the generator power and rotor thrust curves (left) and the design parameters (right). The dashed line corresponds to the hub height wind speed for the precursor ABL simulation used in this study which is situated in region 2 for the this turbine.

provided in Table 1, and the blade pitch signals are shown in Figure 5 over a single Strouhal period. Additionally, a visualization of the streamwise velocity on the hub height plane for each AWM strategies is shown in Figure 1.

The blade pitch fluctuations are designed to induce azimuthal and temporal variations in the blade loading at the wavenumbers and frequencies input to the controller. These variations can be examined through the spectrum of the axial force along the blade span. Following Cheung et al. (2024b), the Fourier representation of the axial force, $F_x$, at a particular radial location, $r$, blade azimuthal angle, $\theta$, and time, $t$, for a given blade is

$$\hat{F}_x(r, \kappa_\theta, \omega) = \int \int F_x(r, \theta, t) e^{-i(\omega t + \kappa_\theta \theta)} d\theta dt. \qquad (2)$$

Given a discrete time series of the axial force at each nodal location along the blade, $F_x(r, t)$, the Fourier coefficients are determined by first approximating the blade azimuthal angle through the mean rotor speed, $\langle \Omega \rangle$, as $\theta(t) \approx \langle \Omega \rangle t + \theta(0)$, allowing equation 2 to be expressed as

$$\hat{F}_x(r, \omega') = \int F_x(r, t) e^{-i\omega' t} dt, \qquad (3)$$

**Table 1.** Summary of cases and AWM parameters.

| Case Name | Pitch Amplitude | Azimuthal Wavenumbers | Strouhal Number | Clocking Angle |
| --- | --- | --- | --- | --- |
| | $(A)$ | $(\kappa_\theta)$ | $(St)$ | $(\phi_{\text{clock}})$ |
| Baseline | N/A | N/A | N/A | N/A |
| Pulse | $1.25°$ | $0$ | $0.3$ | $90°$ |
| Helix | $1.25°$ | $-1$ | $0.3$ | $90°$ |
| Up-and-down | $1.25°$ (each) | $\pm 1$ | $0.3$ | $0°$ |
| Side-to-side | $1.25°$ (each) | $\pm 1$ | $0.3$ | $90°$ |





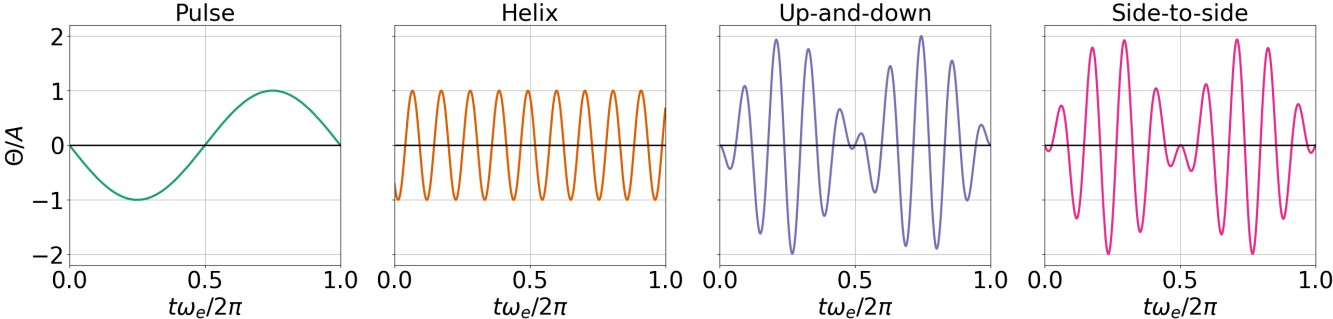

**Figure 5.** Time series of the blade pitch signal for a single blade for each AWM cases, normalized by the pitching amplitude. One Strouhal period is shown based on the excitation frequency $\omega_e$. The black line at $\Theta/A = 0$ corresponds to the baseline blade pitch signal.

where $\omega' = \omega - \kappa_\theta \langle \Omega \rangle$ is an effective frequency. Then the Fourier coefficients, $\hat{F}_x$, are readily determined through a discrete Fourier transform of the blade loading signal in time. In practice, the blade loading signal is not perfectly periodic in time because of the non-uniform inflow, so a windowed Fourier transform is performed instead.

The blade loading spectra indicate that the prescribed blade pitch fluctuations result in the intended modal response in the axial loading for each AWM strategy (see Figure 6). In all cases, the highest periodic loading occurs near the blade-tip at

around 90% of the blade-span. Notably, the $\kappa_\theta = 0$ actuation exhibits significantly higher periodic loading at $St = 0.3$ than the other cases (Figure 6b). An increase in the axial force of over 10% is observed at the $\kappa_\theta = 0$ wavenumber for the pulse actuation across the majority of the blade span, while the other cases show around a 7.5% increase in their respective modes. There are contributions to the axial force from wavenumbers other than those directly forced by the prescribed blade pitch fluctuations, which may result from factors such as the unsteady inflow, the effects of AWM on the turbine induction field, or

other controller specifications. For instance, the baseline case exhibits a slight increase in the $\kappa_\theta = 0$ mode across the outer-half of the blade, which is further attenuated to 2.5% for the up-and-down actuation (Figure 6).

To analyze the corresponding wavenumber and frequency content in the wake, an SPOD analysis is formulated in the following section.

## 2.2 SPOD formulation

Proper orthogonal decomposition (POD), also referred to as principle component analysis or Karhunen–Loéve decomposition, is a common data analysis technique that identifies a set of deterministic modes that optimally represent the energy in a stochastic process. Since its inception by Lumley (1967), several formulations of POD have emerged and been applied to a wide range of applications (Towne et al., 2018). In this paper, the focus is on spectral POD, which identifies the dominant coherent structures in a flow through a set of spatial-temporal modes (Picard and Delville, 2000). This differs from a space-only

POD (Sirovich, 1987; Aubry et al., 1988; Ali et al., 2017) by considering the frequency content of the data, which is particularly useful in the context of AWM for tracking flow structures that are forced at specific Strouhal numbers. Additionally, a polar decomposition is used here to represent spatial correlations as in Citriniti and George (2000), and a Fourier transform is applied





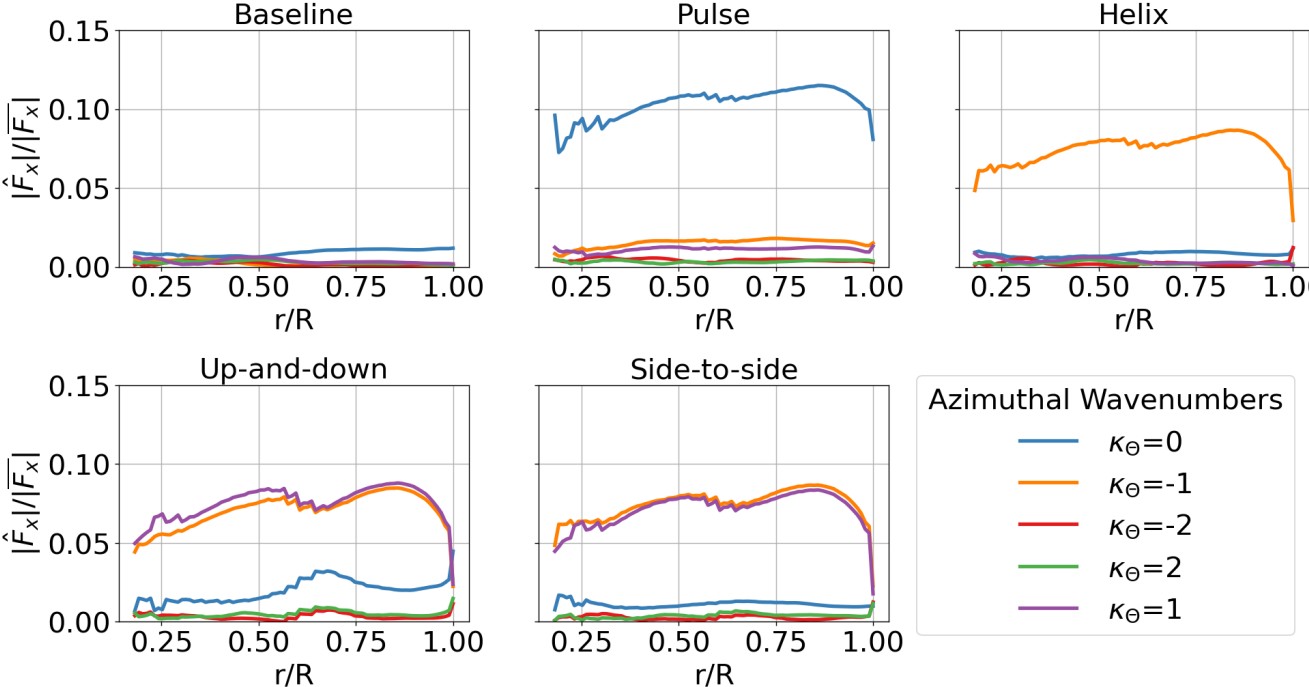

**Figure 6.** Variation in the axial blade loading for each AWM strategy. The individual curves correspond to the magnitude of the Fourier coefficients of the axial force, $|\hat{F}_x|$, at $St = 0.3$ for different azimuthal wavenumbers, $\kappa_\theta$, normalized by the mean axial blade loading, $|\overline{F}_x|$.

in the azimuthal direction so that velocity correlations in the radial direction, $r$, are considered at each azimuthal wavenumber, $\kappa_\theta$, and frequency, $\omega$. This setup allows for the evolution of the forced azimuthal wavenumbers and frequencies that are input
to the turbine controller to be explicitly tracked in the flow, as well as their interactions with other wake structures.

Given a time series of planar velocity field data, the SPOD is defined by the eigenvalues and eigenvectors of the cross-spectral density tensor, $\boldsymbol{S}$, which is the time Fourier-transform of the two-point space-time velocity correlation tensor (Towne et al., 2018). For a statistically stationary flow, the cross-spectral density tensor at a specific frequency, $\omega$, and azimuthal wavenumber, $\kappa_\theta$, can be expressed as $\boldsymbol{S}(r, r') = \langle \hat{\boldsymbol{u}}(r, \kappa_\theta, \omega) \hat{\boldsymbol{u}}^*(r', \kappa_\theta, \omega) \rangle$, where $\hat{\boldsymbol{u}}$ are the velocity Fourier coefficients, $\langle \, \rangle$ is
the expectation operator, and $^*$ represents the complex conjugate of a scalar or the Hermitian transpose of a tensor (Citriniti and George, 2000). In this paper, $\boldsymbol{u}$ denotes the fluctuating velocity field, since SPOD typically operates on a zero-mean stochastic process, and $\boldsymbol{U}$ will be used for the mean velocity field. The eigenvalue problem is then expressed as

$$\int \boldsymbol{S}(r, r', \kappa_\theta, \omega) \boldsymbol{\psi}(r', \kappa_\theta, \omega) dr' = \lambda(\kappa_\theta, \omega) \boldsymbol{\psi}(r, \kappa_\theta, \omega). \tag{4}$$

The solution to Equation 4 is a set of eigenvalues, $\lambda$, and orthogonal eigenvectors, $\boldsymbol{\psi}$. The eigenvectors represent individual
flow structures and the eigenvalues represent the average TKE of flow that is captured by the eigenvectors. Therefore, the eigenvalues can be used to rank the eigenvectors $\boldsymbol{\psi}_1, \boldsymbol{\psi}_2, \boldsymbol{\psi}_3, \cdots$ from most to least energetic across all $\kappa_\theta$ and $\omega$ such that $\lambda_1 \geq \lambda_2 \geq \cdots \geq 0$. In this sense, the velocity Fourier coefficients, $\hat{\boldsymbol{u}}$, can be optimally expanded from the most energetic to the





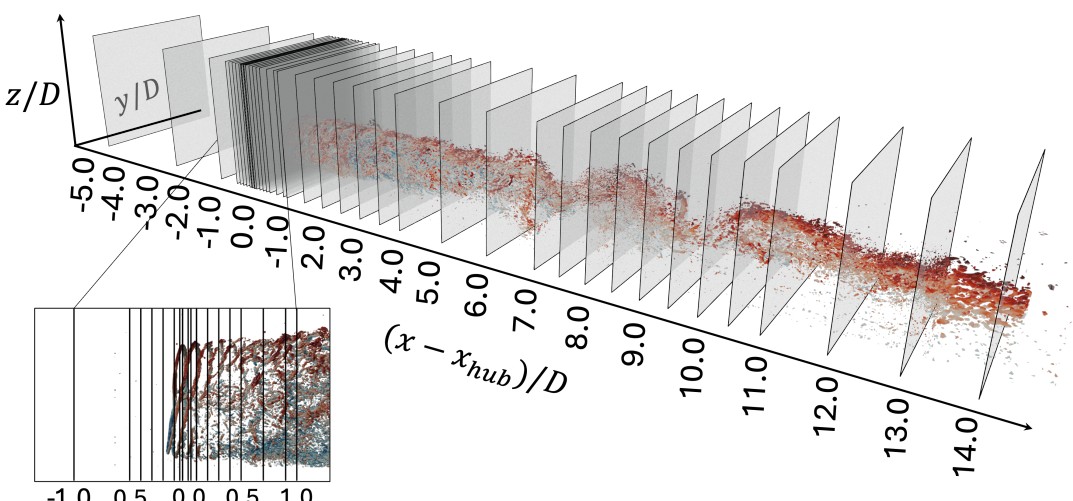

**Figure 7.** Schematic of the cross-flow planes used for the SPOD analysis. Isosurfaces of the baseline wake are also shown, colored by streamwise velocity.

least energetic structures in the flow as

$$\hat{\boldsymbol{u}}(r,\kappa_\theta,\omega) = \sum_{j=1}^{\infty} a_j(\kappa_\theta,\omega)\boldsymbol{\psi}_j(r,\kappa_\theta,\omega) \tag{5}$$

where $a_j(\kappa_\theta,\omega) = \int \hat{\boldsymbol{u}}(r,\kappa_\theta,\omega)\boldsymbol{\psi}_j^*(r,\kappa_\theta,\omega)dr$ is obtained by projecting the velocity Fourier coefficient onto the $j^{\text{th}}$ eigenvector. We can define $\hat{\boldsymbol{u}}_j(r,\kappa_\theta,\omega) \equiv a_j(\kappa_\theta,\omega)\boldsymbol{\psi}_j(r,\kappa_\theta,\omega)$ as the contribution to the velocity Fourier coefficients from the $j$th eigenvector. The real-space velocity field is recovered by performing an inverse Fourier transform in the temporal and azimuthal directions, i.e.,

$$\boldsymbol{u}(r,\theta,t) = \sum_{j=1}^{\infty} \boldsymbol{u}_j(r,\theta,t) = \sum_{j=1}^{\infty} \mathcal{F}_t^{-1}[\mathcal{F}_\theta^{-1}[\hat{\boldsymbol{u}}_j]] \equiv \sum_{j=1}^{\infty} \left[\int\int \hat{\boldsymbol{u}}_j(r,\theta,\omega)e^{i(\kappa_\theta\theta+\omega t)}d\kappa_\theta\, d\omega\right]. \tag{6}$$

In equation 6, $\boldsymbol{u}_j(r,\theta,t)$ denotes the reconstruction of the velocity field in real space from the $j$th SPOD eigenvector. Similarly, the velocity can be reconstructed from the leading $N$ eigenvectors by truncating the summations in (5) and (6). More generally, we denote $\boldsymbol{u}_{\mathcal{J}} = \sum_{j\in\mathcal{J}}\boldsymbol{u}_j(r,\theta,t)$ as the reconstruction of the velocity field from a collection of eigenvectors defined by a given set of integers $\mathcal{J}$. Further details on the general application of SPOD to fluid dynamics, including its connection to Dynamic Mode Decomposition (DMD) and resolvent analysis, are provided by Towne et al. (2018).





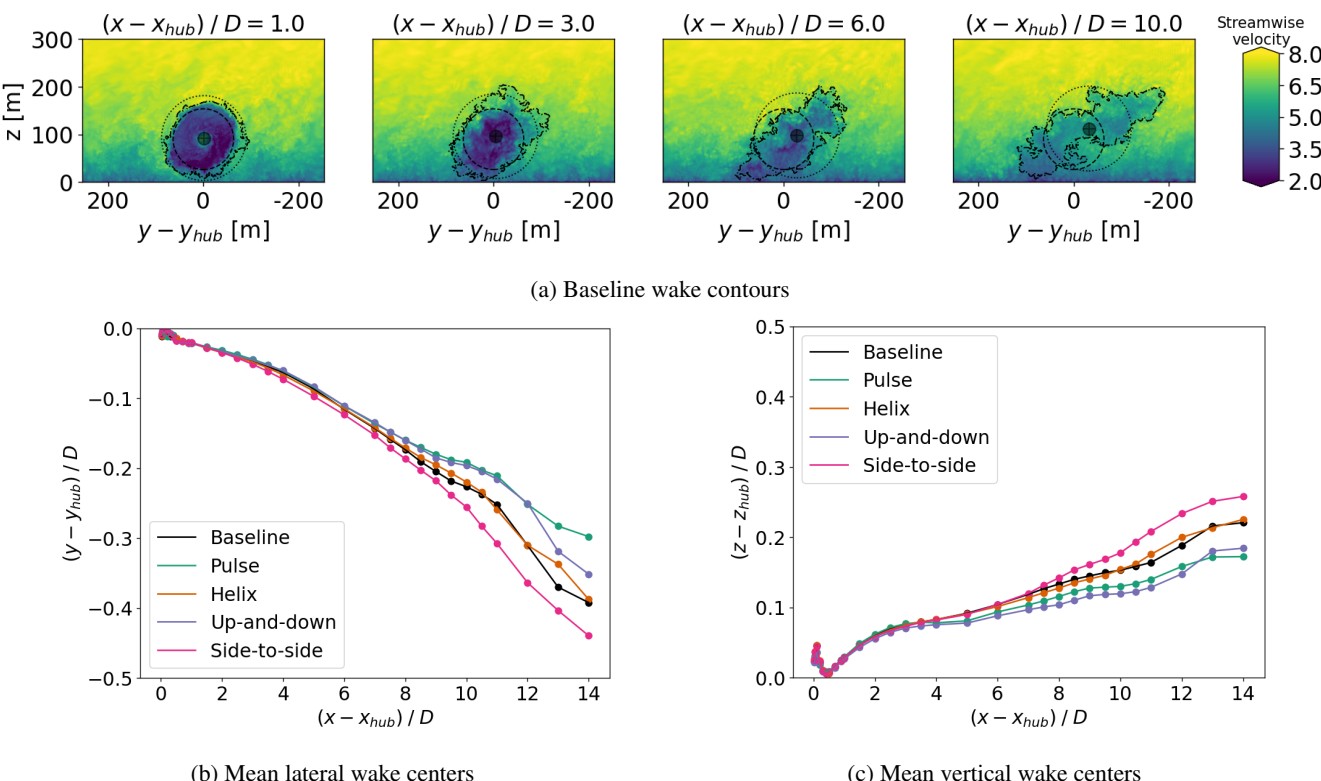

**Figure 8.** (a) Contours of streamwise velocity for the baseline case at $(x - x_{hub})/D = 1, 3, 6$ and 10. Dashed lines indicate the turbine rotor disk, dotted lines indicate a $1.4R$ disk around the wake centers, and dot-dashed lines indicate the wake boundaries identified using the SAMWICH package. The wake centers are also marked. (b) and (c) show the mean lateral and vertical wake centers for each AWM case across the extent of the streamwise domain, respectively.

Given the computational setup in Section 2, SPOD is performed on time series of cross-flow planes extracted from the LES at multiple streamwise locations ranging from $-5 \leq (x - x_{hub})/D \leq 14$ (see Figure 7). These $y$-$z$ planes are sampled at a resolution of 1.25m and a frequency of 2Hz over the 1,100s of simulation time. Each planar time series is divided into nineteen overlapping blocks of data ($N_B = 19$), and the cross-spectral density is approximated from an ensemble average of these realizations for each temporal frequency using a windowed Fourier transform in time. The short-time FFT routine from the SciPy package (Virtanen et al., 2020) is used to perform the padded windowed Fourier transform in time, providing routines for both forward and inverse transforms, allowing the flow field to be reconstructed in real space to machine precision using equation 5 Each block of data corresponds to 128s of simulation time, with a 64s overlap between consecutive blocks. This configuration allows for the resolution of Strouhal numbers down to $St = 0.15$ within each block.

The velocity field is transformed to a polar grid with 256 uniformly spaced points in both the radial, $r \in [0, L_R]$, and azimuthal, $\theta \in [0, 2\pi)$, directions. The radial extent of the flow region considered is taken to be $L_R = 1.4R$, which brings the flow





region to within a 1m clearance from the ground at its lowest point. This radius is found to be sufficient for encompassing the wake near the turbine; however, the wake does spread anisotropically downstream beyond the extent of this radial domain (see Figure 8a). Thus, only the wake dynamics within a $1.4R$ circular region are captured by the SPOD analysis here. This choice is made as the polar representation of the flow field is needed to track the flow structures at the specific azimuthal wavenumbers

forced by turbine controller. However, we note that SPOD could be applied to the full Cartesian data to analyze $y$-$z$ correlations for the entire wake structure, and that, while this would sacrifice the ability to examine specific wavenumbers, it would result in a more efficient basis for representing anisotropic structures in the wake.

For each streamwise location in the wake, the polar coordinates are defined around the center of the wake, rather than a fixed $y - z$ coordinate at all locations. This approach is taken because the turbulence in the wake, including the AWM-induced

flow structures, will follow the mean movements of the wake, and we are interested in tracking their properties downstream. Further, the path the actual wake travels dictates the worst-case position for a downstream turbine (and the best-case scenario for wake control), which further motivates the choice to use the wake position as the centering point for the analyses herein. The wake centers are identified by a velocity deficit weighted average over a $\pi L_R^2$ area using the SAMWICH package (Quon et al., 2020) (see Figure 8a). The lateral and vertical mean wake centers are shown in Figures 8b and 8c, respectively, for

each case in Table 1 across the extent of the streamwise domain. The mean wake generally moves up and to the right (when looking downstream), with the largest deviation from the hub height location occurring for the side-to-side actuation, which moves $0.45D$ to the right and $0.26D$ upward. The wake movement and skewness align with the direction of veer in the ABL. The polar transformation for each AWM case is defined around the wake center for the baseline case. This adjusts the analysis for wake movements as a result of the unsteady inflow to follow the general movement of the dominant flow structures, but

preserves differences in the induced wake movement from the baseline as a result of AWM. Subtraction of the wake centers also adjusts for wake expansion and veer due to mean flow effects. For all streamwise locations upstream of the turbine, the polar representation of the velocity field is centered on the turbine hub height.

All three cylindrical velocity field components, $\boldsymbol{u} = (u_x, u_r, u_\theta)$, are included in the SPOD analysis of the cross spectral density tensor. This approach provides the most comprehensive description of the coherent structures in the flow because it

contains correlations between all components of the velocity field. However, the eigenvalues and eigenvectors of the components of the trace of $\boldsymbol{S}$ can also be used to identify the response of each velocity component to AWM individually (see Appendix A). In this case $S_{11}$, $S_{22}$, and $S_{33}$ are identified with the streamwise, radial, and azimuthal velocity components, respectively. This is useful for discerning which components of the flow are contributing the most TKE to a given flow structure. For both $\boldsymbol{S}$ and its diagonal elements, the discrete eigenvalue problem is solved using a low-rank SVD solver, which results

in $N_B$ eigenvalues and eigenvectors for each $\kappa_\theta$ and $\omega$. More details on the solution to the discrete eigenvalue problem are provided in Appendix B.





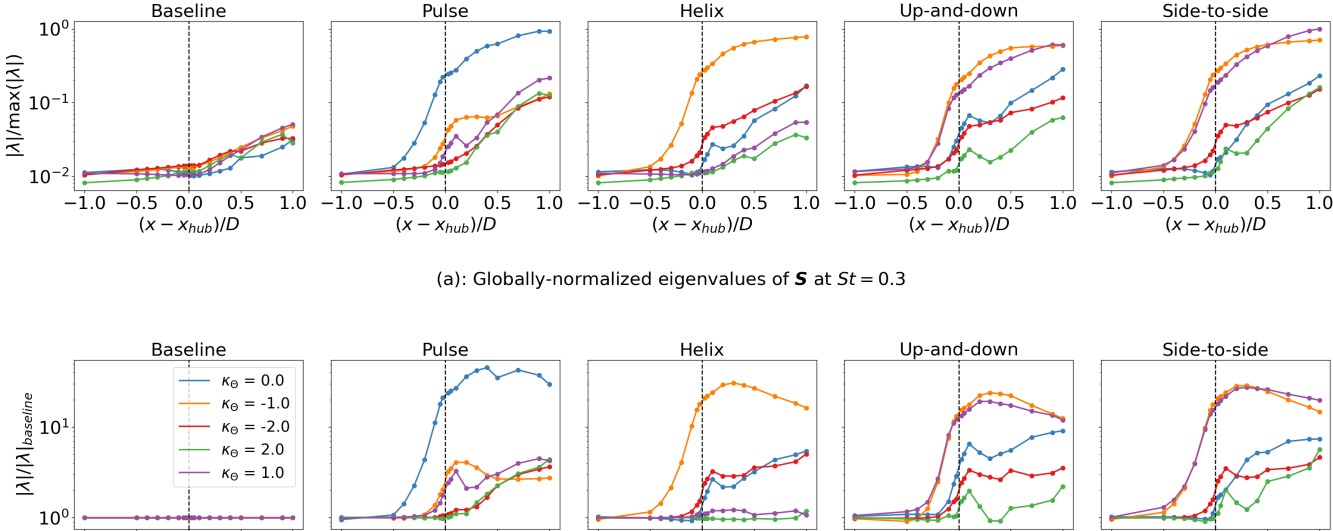

**Figure 9.** Eigenvalues of $\mathbf{S}$ at $St = 0.3$ near the turbine ($|(x - x_{hub})/D| \leq 1$) for the azimuthal wavenumbers $\kappa_\theta = 0$ (—), $\kappa_\theta = 1$ (—), $\kappa_\theta = -1$ (—), $\kappa_\theta = 2$ (—), and $\kappa_\theta = -2$ (—). (a) Each eigenvalue is normalized by the global maximum eigenvalue across all AWM cases and streamwise locations. (b) The eigenvalues for each wavenumber and streamwise location are normalized by the corresponding eigenvalues in the baseline case.

## 3  Results

In this section, the wake mixing dynamics and efficacy of each AWM strategy are explored through the SPOD analysis. The energetic flow structures are tracked throughout the wake in Section 3.1 and connected to turbulent entrainment statistics in Section 3.2. The practical benefits of AWM are also demonstrated for a two-turbine array in Section 3.3.

### 3.1  SPOD results

The SPOD eigenvalues are used to track the dominant flow structures throughout the streamwise domain. The results primarily focus on the azimuthal wavenumbers $\kappa_\theta = 0, \pm 1$, and $\pm 2$ at $St = 0.3$ as this range encompasses the forced coherent structures for each AWM case, as well as other large-scale structures that get excited downstream in the wake. The eigenvalues for the baseline case are also analyzed at these wavenumbers and frequency, which represent the energy in the inherent flow structures in the non-actuated wake resulting from standard turbine operations. Both the absolute values of the eigenvalues (Figures 9a and 10a) and the baseline-normalized eigenvalues (Figures 9b and 10b) are discussed, which quantify the dominant coherent structures in the wake for each control strategy and the effectiveness of AWM at exciting structures in the wake over the baseline, respectively. Comparisons in the modal behavior for each AWM case offer insights into their relative performance,



WIND
ENERGY
SCIENCE
DISCUSSIONS
eawe
european academy of wind energy
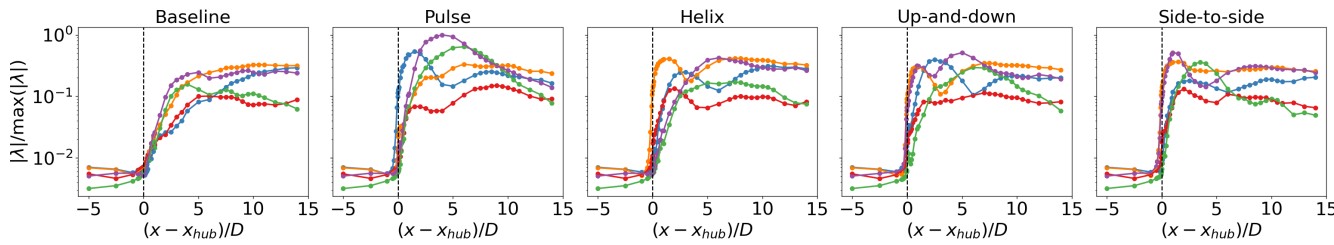

(a): Globally-normalized eigenvalues of **S** at $St = 0.3$

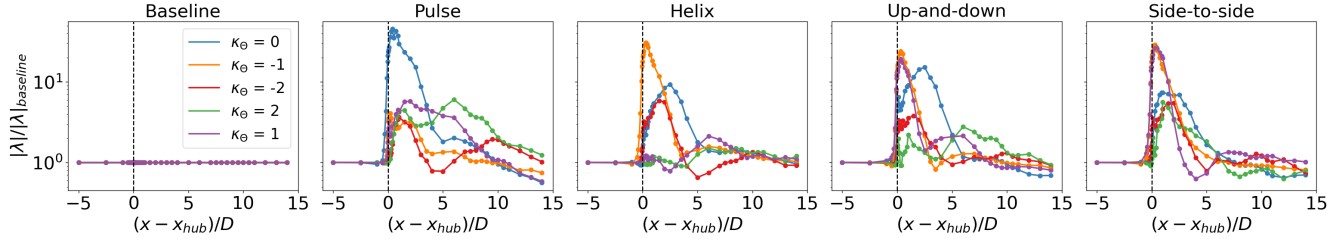

(b): Baseline-normalized eigenvalues of **S** at $St = 0.3$

**Figure 10.** Eigenvalues of **S** at $St = 0.3$ for $-5 \leq (x - x_{hub})/D \leq 14$ for the azimuthal wavenumbers $wkappa_\theta = 0$ (—), $\kappa_\theta = 1$ (—), $\kappa_\theta = -1$ (—), $\kappa_\theta = 2$ (—), and $\kappa_\theta = -2$ (—). (a) Each eigenvalue is normalized by the global maximum eigenvalue across all AWM cases and streamwise locations. (b) The eigenvalues for each wavenumber and streamwise location are normalized by the corresponding eigenvalues in the baseline case.

which are strengthened further in Sections 3.2 and 3.3. In addition to eigenvalues, the reconstruction of the velocity field from the leading SPOD eigenvectors is used to show the leading flow structures in the wake (Figure 11) and the induction field (Figure 13).

In the baseline case, a small distinction between large-scale flow structures originates in the induction field ahead of the turbine, which grows significantly in the wake (see Figures 9a and 10a). The $\kappa_\theta = -1$ mode is the dominant flow structure in 265 the immediate near-wake of the turbine as well as in the far-wake, while the $\kappa_\theta = 1$ mode is the leading structure between $0.5D$ and $5D$ downstream of the turbine. Together, the $\pm 1$ modes result in a movement of the wake, as well as a swirl component if the energy in these modes is imbalanced. SPOD decouples these competing rotational effects in the wake, allowing them to be examined independently. There are several factors contributing to the swirl of the baseline wake. The clockwise rotation of the turbine blades, for instance, imparts a counter-clockwise swirl in the mean wake, which may be contributing to the large 270 $\kappa_\theta = -1$ mode. Similarly, the positive veer in this ABL introduces a horizontal shear to the wake, resulting in a large-scale counter-clockwise swirl, which may be contributing to the $\kappa_\theta = 1$ mode. Other prominent energetic structures in baseline wake include the $\kappa_\theta = 2$ mode in the near wake region, and the $\kappa_\theta = 0$ mode in the far wake.

For each AWM case, the largest increase in eigenvalues over the baseline occurs in the near-wake region between $0.25D$ and $1D$ behind the turbine, corresponding to the azimuthal wavenumber directly forced by the dynamic blade pitch settings



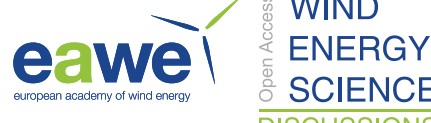

(a): Leading SPOD modes for $St = 0.3$ at $(x - x_{hub})/D = 0.1$

(b): Leading SPOD modes for $St = 0.3$ at $(x - x_{hub})/D = 1.0$

(c): Leading SPOD modes for $St = 0.3$ at $(x - x_{hub})/D = 3.0$

(d): Leading SPOD modes for $St = 0.3$ at $(x - x_{hub})/D = 6.0$

**Figure 11.** Dominant flow structures at four streamwise locations in the wake. The reconstructed fluctuating velocity field, $\boldsymbol{u}_{\mathcal{J}}(r,\theta)$ is shown at a single instance in time using the SPOD eigenvectors that correspond to the leading eigenvalues for $St = 0.3$. For each case, the reconstructed velocity field is normalized by its maximum value in the polar domain at that specific time. For the pulse and helix cases, one SPOD mode is included in the reconstruction, i.e., $\mathcal{J} = \{1 \mid \omega U_{\mathrm{inf}}/D = 0.3\}$. For the up-and-down and side-to-side cases two SPOD modes are included in the flow reconstruction, i.e., $\mathcal{J} = \{1, 2 \mid \omega U_{\mathrm{inf}}/D = 0.3\}$. The colored contours show normalized streamwise velocity and the arrows denote the in-plane velocities.





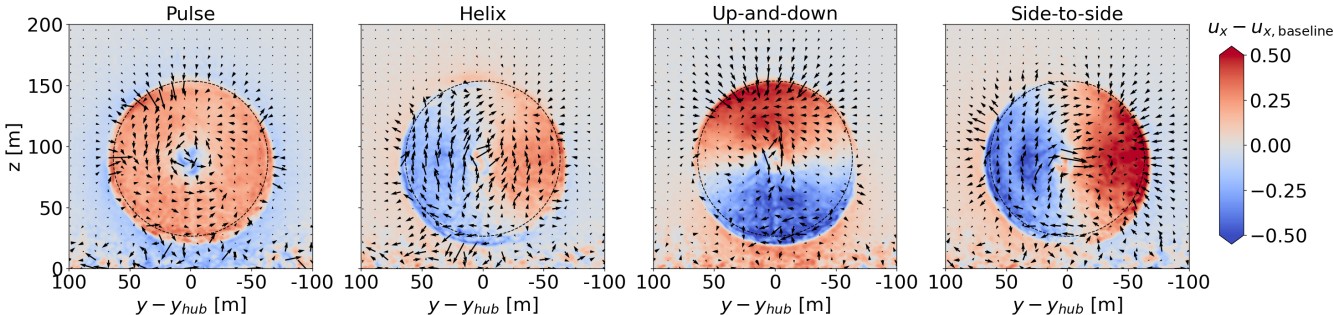

**Figure 12.** Phase-averaged, baseline-subtracted velocity fields at $(x - x_{hub})/D = 0.1$. Phase averaging within the Strouhal cycle is performed with a $10°$ bin centered on $0°$. After the phase averaging is performed for all cases, both the streamwise and lateral components of velocity are baseline-subtracted to yield the colored contours and the vector fields, respectively. The dashed lines outlines the turbine rotor disk.

i.e., $\kappa_\theta = 0$ for the pulse, $\kappa_\theta = -1$ for the helix, and $\kappa_\theta = \pm 1$ for the up-and-down and side-to-side actuations (see Figures 9b and 10b). The SPOD analysis therefore confirms that the azimuthal wavenumber and frequency inputs to the turbine controller lead to the intended flow response directly downstream of the turbine. The streamwise velocity component contributes the most to the TKE of the coherent structures in the near wake region, although a similar modal response is induced in the radial and azimuthal components (see Appendix A). This is expected as the most observable effect of the blade pitch fluctuations

are on the axial loading. It is also readily apparent from the leading SPOD mode shapes that the intended flow perturbations are imparted on the wake for each AWM case (see Figure 11a). The primary flow perturbation in the near wake for the pulse actuation is axisymmetric and oscillates at the Strouhal period. The coherent structures for the other AWM strategies are characterized by two regions of low- and high-speed flow, which rotate about the wake center for the helix case and oscillate at the Strouhal period for the mixed-mode forcings depending on the clocking angle. To connect the SPOD eigenvectors to more

familiar flow quantities, we note that these flow patterns can also be observed in the region near the rotor by carefully phase averaging the real-space flow. In Figure 12, the phase-averaged baseline-subtracted velocity field is shown for each AWM case. The leading SPOD modes are clearly apparent in the phase-averaged fields, demonstrating that the flow patterns identified by SPOD are indeed the dominant coherent structures in the wake.

The eigenvalues of the forced azimuthal wavenumber(s) begin to increase from the baseline values around $0.5D$ upstream

of the turbine for each AWM strategy, and grow rapidly through the induction field to reach a value over an order of magnitude larger than the baseline at the turbine location (see Figure 9). AWM therefore modifies the induction field with similar spectral characteristics as those imparted on the wake region directly behind the turbine. However, unlike in the wake, the coherent structures at $St = 0.3$ are not the dominant energy containing modes in the induction field, which instead occur near the blade-passing frequency ($St \approx 7.5$) for each AWM case (see Figure 13a). Nonetheless, SPOD allows these structures to be

decoupled, and the focus here is on modes at the excitation frequency (Figure 13b). The azimuthal and temporal modifications to the turbine induction field translate into variations in the axial force along the blade (compare Figures 6 and 9), which are subsequently imparted to the downstream wake. Notably, the $\kappa_\theta = 0$ in the near-wake region of the pulse case exhibits a larger



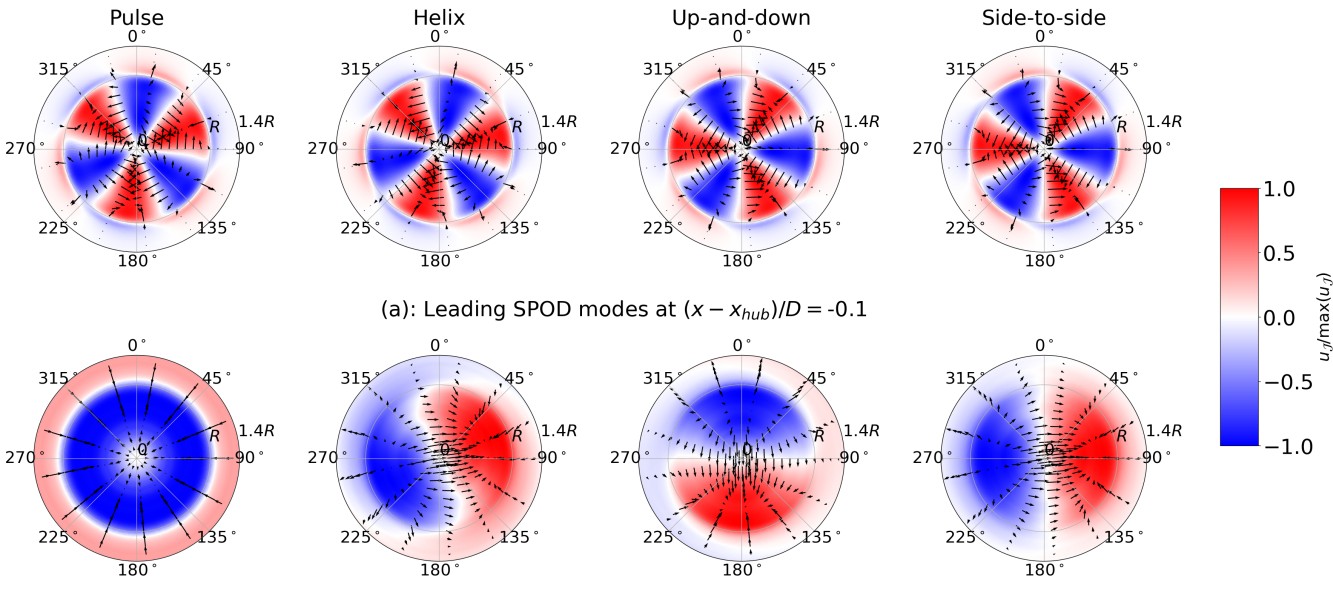

(a): Leading SPOD modes at $(x - x_{hub})/D = -0.1$

(b): Leading SPOD modes for $St = 0.3$ at $(x - x_{hub})/D = -0.1$

**Figure 13.** Dominant flow structures in the induction field. The reconstructed fluctuating velocity field, $\boldsymbol{u}_{\mathcal{J}}(r, \theta, t)$ is shown at a single instance in time. For each case, the reconstructed velocity field is normalized by its maximum value in the polar domain at that specific time. For the pulse and helix case, one SPOD mode is included in the reconstruction, whereas for the up-and-down and side-to-side cases two SPOD modes are included in the flow reconstruction. In (a), the leading SPOD eigenvectors across all wavenumbers and frequencies are used, i.e., $\mathcal{J} = \{1\}$ for the pulse and helix methods, and $\mathcal{J} = \{1, 2\}$ for up-and-down and side-to-side methods. In (b), the leading SPOD eigenvectors across all wavenumbers at $St = 0.3$ are used, i.e., $\mathcal{J} = \{1 \mid \omega U_{\text{inf}}/D = 0.3\}$ for the pulse and helix method and $\mathcal{J} = \{1, 2 \mid \omega U_{\text{inf}}/D = 0.3\}$ for the up-and-down and side-to-side method. The colored contours show normalized streamwise velocity and the arrows denote the in-plane velocities.

increase in energy over the baseline compared to the forced wavenumbers associated with the other AWM strategies. This observation aligns with the larger variations in blade loading seen with the pulse method compared to the other cases (see

Figure 6). The structure of the dominant modes in the induction field may provide insight into why there is increased blade loading observed with pulse actuation relative to the other AWM strategies. For the non-axisymmetric forcings, the in-plane velocity field in the induction zone is seen to divert the flow from the low- to high-speed region of the streamwise velocity (see Figure 11b). This suggests that the individual pitching of the turbine blades creates a non-uniform blockage in the flow, leading to flow diversion around the region of higher blockage and consequently reducing the periodic axial blade force that

actuates the wake modes. However, in the pulse case, there is no apparent flow diversion because all the blades are collectively pitched. This increase in blade loading for the pulse case is seemingly important as it translates to the largest modal energy in the near-wake region which, as shown below, facilitates the most effective mixing of the downstream wake.



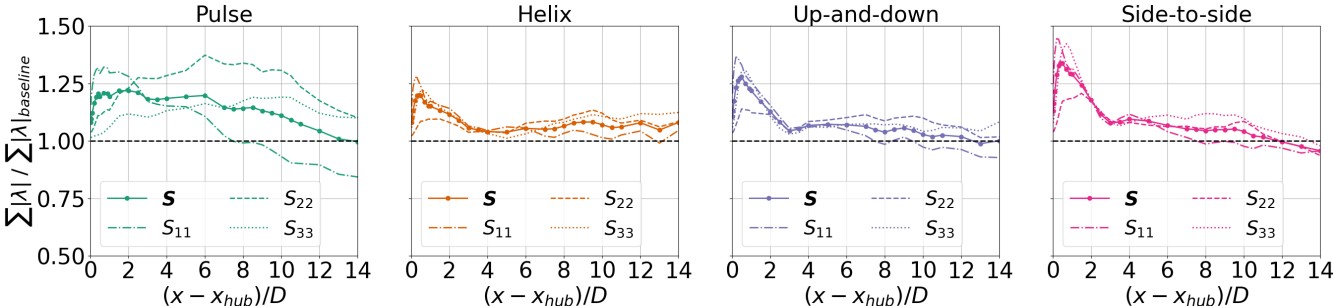

**Figure 14.** Baseline normalized TKE in the wake computed as the sum of the magnitude of eigenvalues of $\boldsymbol{S}$ across all wavenumbers and frequencies. The baseline normalized contribution to TKE from the streamwise, radial, and azimuthal velocity components is also shown and computed from the eigenvalues of $S_{11}$, $S_{22}$, and $S_{33}$, respectively.

As the wake evolves downstream, nonlinear and mean flow interactions excite flow structures beyond those directly forced by the turbine controller. For the non-axisymmetric forcing strategies, there is a notable increase in the $\kappa_\theta = 0$ wavenumber relative to the baseline between $2 \leq (x - x_{hub})/D \leq 5$, which dominates the baseline-normalized eigenvalues in this region (Figure 10b). For the pulse case, the increase over the baseline is sustained by the $\kappa_\theta = 1$ and 2 wavenumbers between $3 \leq (x - x_{hub})/D \leq 10$. By 10 to 14D downstream, the $\kappa_\theta = -1$ mode is dominant in all cases, but the modal energy in the large scales has generally returned to the baseline levels for all of the AWM strategies. However, we note that increases in pitching amplitude or adjustments to the Strouhal number may extend the duration of AWM effects. Moreover, it is important to highlight that, although the modal energy relative to the baseline wake generally decreases downstream for the AWM cases, the modal energy in the baseline wake itself typically increases downstream (Figure 10a). Consequently, the flow structure with the largest modal energy for the AWM cases may not be located in the immediate near-wake of the turbine, despite the largest increases over the baseline occurring in that region. It is clear from the SPOD modes, for instance, that the dominant flow structure, in terms of the absolute magnitude of the eigenvalues, changes as the wake evolves downstream (see Figures 11). The largest absolute eigenvalue across all AWM cases and streamwise locations occurs for the $\kappa_\theta = 1$ wavenumbers for the pulse actuation $4D$ downstream (see Figure 10a). In fact, for all AWM cases, the largest eigenvalues between 5 and 6D downstream is for the $\kappa_\theta = 1$ wavenumber. The energy in this mode is primarily driven by the in-plane velocity components ($u_r$ in this case), which is generally true of the modal energy increases over the baseline by $4D$ to $5D$ downstream (see Appendix A). Notably, the $\kappa_\theta = 1$ mode is also the dominant structure in the near-wake region of the baseline case, which suggests that the effectiveness of AWM strategies at increasing modal energy in the wake is related to their ability to excite the natural flow structures in the non-actuated wake. This appears to be the case for higher harmonics in the flow as well, such as the $\kappa_\theta = 2$ mode for the pulse and side-to-side actuations (see Figure 10b).

The flow response in the wake between the two mixed-mode forcing strategies differs significantly even though both strategies excite the $\kappa_\theta = \pm 1$ modes. This behavior highlights that the different clocking angles cause the flow structure to interact differently with the ABL, including variations in shear, stratification, turbulence, and ground effect. By $3D$ to $6D$ downstream,



WIND
ENERGY
SCIENCE
DISCUSSIONS

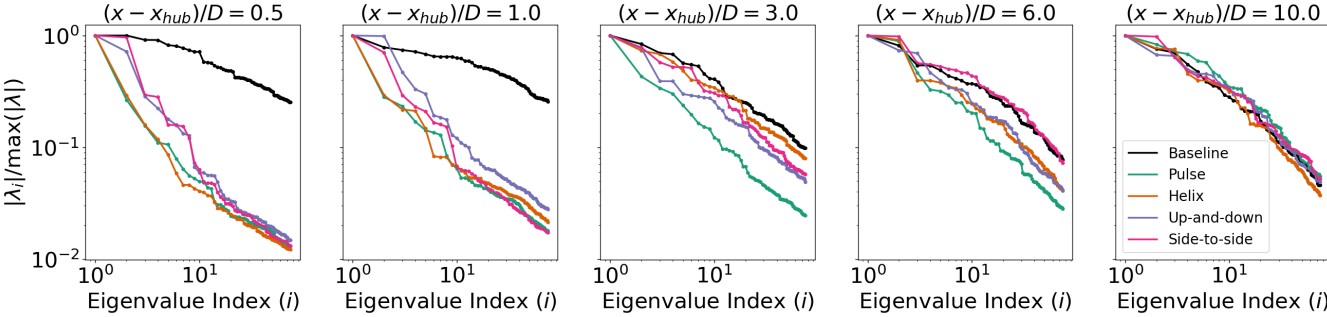

**Figure 15.** The eigenvalue spectra for each AWM strategy are shown at five streamwise locations in the wake. The leading 75 eigenvalues across all azimuthal wavenumbers and frequencies are included. Each spectrum is normalized by its leading eigenvalue, $|\lambda_1|$ to emphasize the relative decay in eigenvalues between cases.

the up-and-down strategy strongly forces the $\kappa_\theta = 0$ and $1$ modes, while the side-to-side strategy forces the $\kappa_\theta = 2$ mode (Figure 10). Interestingly, the $\kappa_\theta = 1$ and $-1$ modes have nearly identical growth rates in the induction field ahead of the turbine for these cases, but, directly behind the turbine, $\kappa_\theta = -1$ emerges as the dominant flow structure for both cases. This suggests that the $\kappa_\theta = -1$ mode is enhanced by the counter-clockwise vortex rings generated behind the turbine, which has also been linked to the performance differences between the counter-clockwise and clockwise helix strategies (Coquelet et al., 2024).

The total TKE in the wake at each streamwise location can be computed by summing the entire set of eigenvalues across all wavenumbers and frequencies (see Figure 14). In the near wake, the up-and-down and side-to-side forcings have the largest TKE increase over the baseline, because two modes are forced at the same pitching amplitude as the single modes in the pulse and helix case. However, by $3D$ downstream, the increase in TKE over the baseline drops to around 10% for the non-axisymmetric forcings, while the pulse case maintains around a 15-20% increase until $9D$ downstream. This increase for the pulse actuation is driven by TKE in the radial velocity (see $S_{22}$ in Figure 14). It is important to note that the goal of AWM is to increase TKE in the large scale coherent structures of the wake that entrain the freestream momentum, not necessarily to increase wake TKE in general, which can impact blade loading on downstream turbines. To understand the separation of scales in the wake, the streamwise evolution of the eigenvalue spectra is reported in Figure 15. A large eigenvalue decay is observed in the near wake region ($(x - x_{hub})/D < 3$), indicating that the flow is well represented by the leading 1-2 SPOD modes. In this region, the large-scale coherent structures are dominating the unsteadiness of the flow and account for the majority of wake TKE. However, by $10D$ downstream, the distribution of modal energy for the AWM cases is similar to the turbulence in the baseline wake.

## 3.2 Wake mixing and turbulent entrainment dynamics

The practical benefit of increased modal energy in the wake is additional entrainment of mean momentum by the coherent wake structures. This is the primary mechanism by which AWM enhances wake recovery and where the advantage of AWM lies over other control strategies. Entrainment is quantified here using the negative radial shear stress flux, $\mathcal{T} = -U_x \overline{u_x u_r}$, where $\overline{\phantom{x}}$

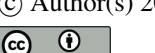



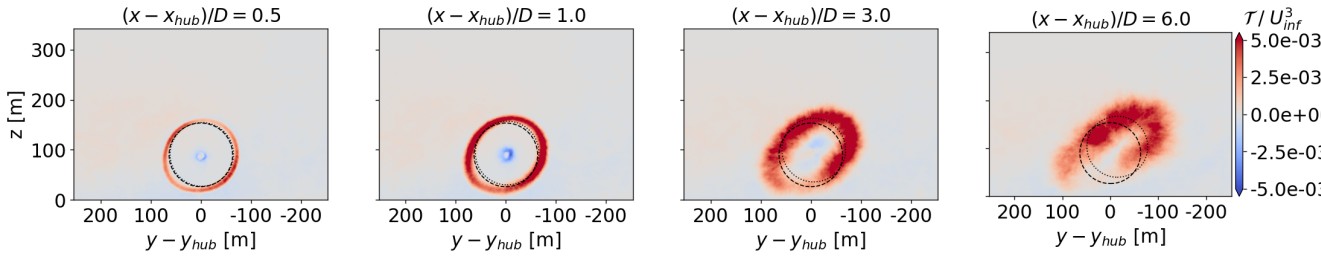

**Figure 16.** Contours of the radial shear stress flux, $\mathcal{T}$, for the baseline case at four different streamwise locations. The dashed line corresponds to the rotor disk centered at the hub height location and the dotted line corresponds to the rotor disk centered around the wake centers.

denotes the time average of a fluctuating quantity. This term quantifies the radial turbulent transport of mean streamwise kinetic energy, which Lebron et al. (2012) demonstrated is the dominant contributor to wake recovery for a single, isolated turbine

wake. Positive values of $\mathcal{T}$ indicate a gain of mean flow into the wake due to turbulent transport, whereas negative values of $\mathcal{T}$ indicate a loss of mean flow.

The radial shear stress flux of the wake for the baseline case is shown in Figure 16. Mean flow is primarily entrained through the boundary of the wake, which aligns with the rotor-disk in the near wake region but deforms anisotropically downstream. Veer, in particular, skews the wake significantly downstream, which in turn affects the distribution of $\mathcal{T}$. A scalar measure of

mean flow entrainment, $\tau = \frac{1}{2\pi} \int_0^{2\pi} \mathcal{T}(r = D/2, \theta)d\theta$, is obtained by azimuthally averaging $\mathcal{T}$ around the circumference of a rotor disk (see Figure 17a). Wake recovery mechanisms are analyzed here in terms of a rotor disk centered around the wake centers at each streamwise location to track the entrainment properties of the wake structures downstream. This formulation is consistent with the SPOD analysis, which is computed around the wake centers.

All AWM strategies improve the net radial shear stress flux over the baseline throughout most of the streamwise domain

(see Figure 17a), with the largest increases occurring between $2D$ and $4D$ behind the turbine. However, this improvement is not homogeneous around the rotor disk, and there are points where mean flow is even lost compared to the baseline due to the AWM pitch actuation (see Figure 18). This is a departure from the canonical case analyzed by Cheung et al. (2024b), further highlighting the impact of ABL characteristics on wake mixing dynamics including wind veer, shear, and stratification. Notably, the largest increase in the net mean flow entrainment is for the pulse actuation, which sustains an increase over the

baseline between $3D$ and $10D$ downstream (see Figure 17a), whereas the other AWM cases return to the baseline levels in this region. This behavior is similar to the downstream evolution of the eigenvalues for the pulse case, which also sustained an increase over the baseline in this region (Figure 10a). The modal structure of the forcing is visible in the contours of $\mathcal{T}$ within a turbine diameter downstream, where positive values are approximately axisymmetric for the pulse case but depend on the clocking angle for the up-and-down and side-to-side actuations (see Figure 18); a more sophisticated analysis is needed to

discern modal behavior farther downstream.

While Figure 17a provides an overview of the total radial shear-stress flux, SPOD facilitates more detailed analysis of the contributions to this flux. The contribution from each flow structure in the wake to turbulent entrainment can be quantified





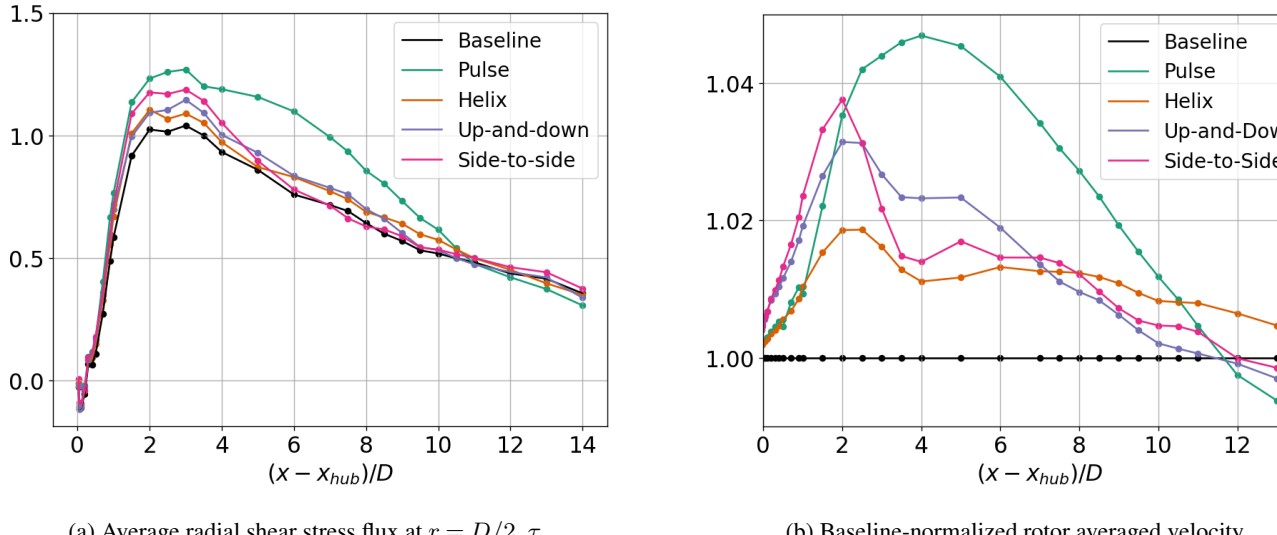

(a) Average radial shear stress flux at $r = D/2$, $\tau$.

(b) Baseline-normalized rotor averaged velocity.

**Figure 17.** (a) Circumferential average of the radial shear stress flux around a rotor disk centered at the wake center. (b) Rotor averaged velocity for each AWM case, normalized by the baseline value. The velocity is averaged over a rotor disk centered around the wake center.

by decomposing the radial shear stress flux into SPOD modes. Specifically, we can define the contribution to $\mathcal{T}$ from the $j$th SPOD eigenvectors as

$$\mathcal{T}_{j,j}(r,\theta) = -U_x \overline{u_{x,j} u_{r,j}}, \tag{7}$$


where $u_{x,j}$ and $u_{r,j}$ are the streamwise and radial components of $\boldsymbol{u}_j$ (see equation 6). A full decomposition of $\mathcal{T}$ is obtained by considering streamwise and radial velocity correlations between all pairs of SPOD eigenvectors, namely $\mathcal{T}_{j,k}(r,\theta) = -U_x \overline{u_{x,j} u_{r,k}}$. However, the focus here is on the case when $j = k$, which is found to dominate $\mathcal{T}_{j,k}$ for all wavenumbers and frequencies of interest. This is expected as the AWM forcings induce a similar modal response in the streamwise and radial

velocity components (see Appendix A). Similarly to $\mathcal{T}$, the net mean flow entrainment into the rotor disk from $\mathcal{T}_{j,j}$ can be quantified by $\tau_{j,j} = \frac{1}{2\pi} \int_0^{2\pi} \mathcal{T}_{j,j}(r = D/2, \theta) d\theta$.

This decomposition enables a quantitative comparison between the modal contributions to turbulent entrainment, $\tau_{j,j}$, and modal TKE, $|\lambda_j|$, in the wake. For each AWM case, the wavenumbers directly forced by the blade pitch actuations entrain the most momentum directly behind the turbine (see Figure 19); however, this is not necessarily the case throughout the

entire wake. Turbulent entrainment for the pulse actuation is driven by the $\kappa_\theta = 1$ and 2 wavenumbers between $3D$ and $10D$ downstream (Figure 19), explaining the large aforementioned increase in $\tau$ over the baseline for the pulse case in this region. (Figure 17a). However, the contribution from the $\kappa_\theta = 0$ mode drops below that of the baseline for the pulse case by $4D$ downstream. The $\kappa_\theta = 1$ and 2 wavenumbers are the dominant flow structures in the near wake of the baseline case, in terms of both TKE and entrainment. This suggests that optimizing AWM depends on forcing the natural coherent structures in the non-

actuated wake, either directly through the turbine controller, or indirectly through triadic wavenumber interactions (Waleffe,







**Figure 18.** Contours of the baseline-subtracted radial shear stress flux for each AWM case. The dashed line corresponds to the rotor disk centered at the hub height location and the dotted line corresponds to the rotor disk centered around the wake centers.

1992) in the wake and interactions with other ABL processes. Similarly, the $\kappa_\theta = 1$ and 2 modes contribute significantly to entrainment for both mixed-mode forcing strategies, however, their entrainment characteristics in the wake differ. In the up-and-down case, the $\kappa_\theta = 1$ and 2 modes contribute the most to entrainment between $4D$ and $10D$ downstream following the excitation of the 0 mode in the near-wake region. In contrast, for the side-to-side case, the $\kappa_\theta = 2$ mode entrains the most

momentum in the near wake while the 1 mode dominants in the far wake. Finally, even though the counter-clockwise helix method is used here to excite the $\kappa_\theta = -1$ mode, the largest contribution to entrainment in the wake is from the $\kappa_\theta = 1$ mode between $5D$ and $7D$ downstream.

Given the prominence of the $\kappa_\theta = 1$ structure in this flow, it would be worthwhile to investigate a clockwise helix method in this context, as it would directly excite this mode. Although we do not generally expect the clockwise helix method to





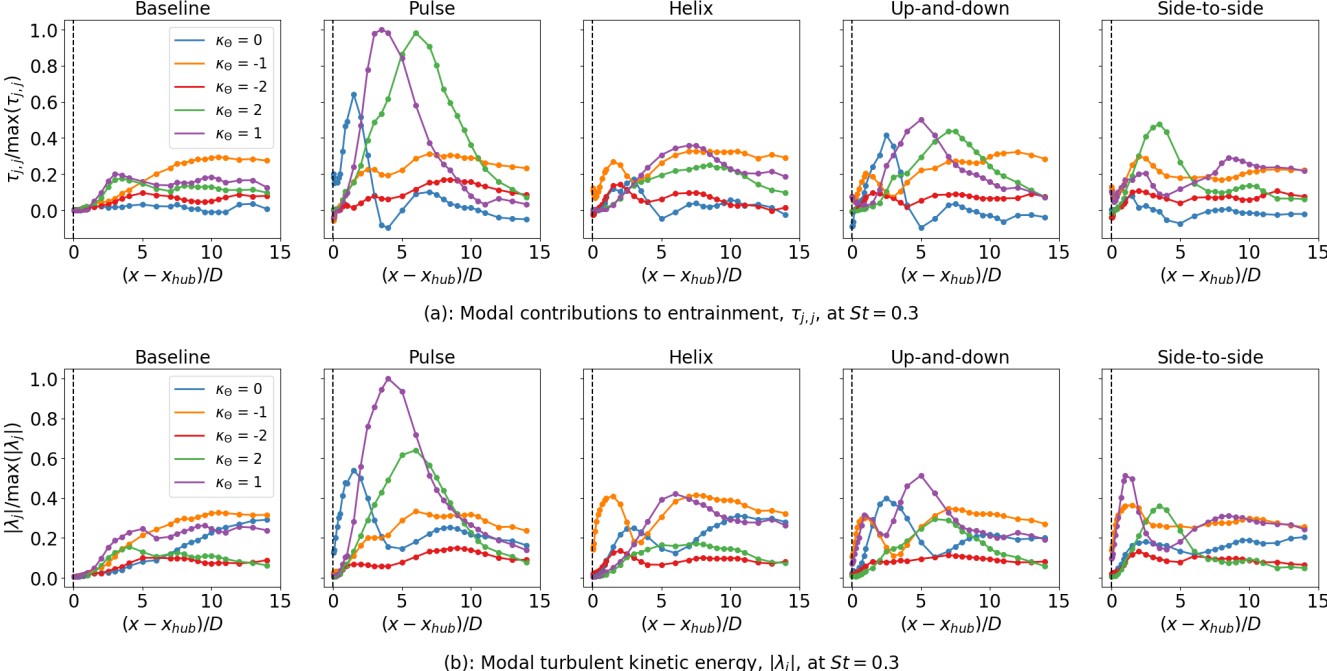

**Figure 19.** (a) Azimuthally averaged contribution to the radial shear stress flux, $\tau_{j,j}$, and (b) TKE, $|\lambda_j|$, in the wake from the leading SPOD eigenvectors at $St = 0.3$ and $\kappa_\theta = 0, \pm 1$, and $\pm 2$. Each AWM case is shown in addition to the baseline. The values in (a) and (b) are normalized by the maximum values of $\tau_{j,j}$ and $|\lambda_j|$ across all AWM strategies and streamwise locations, respectively.

outperform the counter-clockwise helix (Coquelet et al., 2024), it may be that the presence of veer enhances the former's effect by introducing a clockwise swirl in the wake. It is worth noting, however, that while the two mixed-mode cases do force the $\kappa_\theta = -1$ mode directly, the pulse method enhances this mode the most. Strategies that force higher harmonics may also be worth investigating, as the $\kappa_\theta = 2$ mode contributes significantly to entrainment for the pulse, up-and-down, and side-to-side strategies, although increasing $|\kappa_\theta|$ beyond 1 in the turbine controller may introduce prohibitive oscillations in the pitch of the turbine blades.

Across a wider range of turbulent scales and frequencies, it is generally found that the dominant coherent structure in terms of TKE are also responsible for entraining the most mean flow (see Figure 20). However, $\tau_{j,j}$ does not strictly decrease with the eigenvalue index $j$ like $|\lambda_j|$ does; there are several instances across streamwise locations and AWM cases where the flow structure that entrains the most momentum is not the one with the largest modal energy, although it typically falls within the leading 5-10 eigenvalues (Figure 20). This relationship between modal TKE and entrainment breaks down for the smaller scales of turbulence. Once the eigenvalue spectrum has decayed by roughly an order of magnitude, small scale turbulent structures even begin to entrain mean flow out of the wake as evident by the negative values of $\tau_{j,j}$ for large eigenvalues indices, $j$ (see Figure 20).



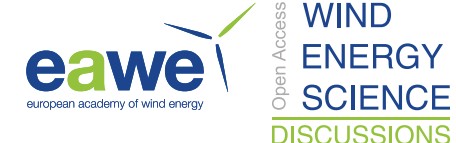

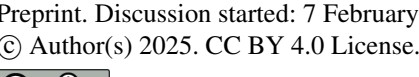

**Figure 20.** Energy and entrainment spectra defined by $|\lambda_j|$ and $\tau_{j,j}$, respectively, for the leading 75 eigenvalues at $(x - x_{hub})/D = 1$, 3, and 6. The spectra are normalized on the maximum value of $|\lambda_j|$ and $\tau_{j,j}$ for each AWM case and streamwise location. Note that $\tau_{j,j}$ is not a strictly positive quantity like $|\lambda_j|$, so large discontinuities in the log-spectra plots are observed, representative of scales contributing to the loss of mean flow in the wake.

The increases in the radial shear stress flux over the baseline case as described above result in faster velocity recovery in the wake (see Figures 21 and 22). Similar to $\mathcal{T}$, the increases in mean streamwise velocity, $U_x$, are not distributed uniformly around the rotor disk; however, all AWM strategies lead to a net increase in rotor-averaged streamwise velocity over the baseline across most of the streamwise domain (see Figure 17b). The largest increase occurs with the pulse actuation, which shows over a 4% increase from the baseline between $2D$ and $6D$ downstream, while the other AWM strategies exhibit increases of 1% to 2.5% in this region. Beyond $12D$ downstream, the trend reverses, and the rotor-averaged velocity for the pulse actuation falls the furthest below the baseline value. However, at such far distances downstream, there may still be a benefit from AWM in terms of wake recovery if averaging over a larger volume. Notably, the helix forcing does sustain an increase in rotor-averaged





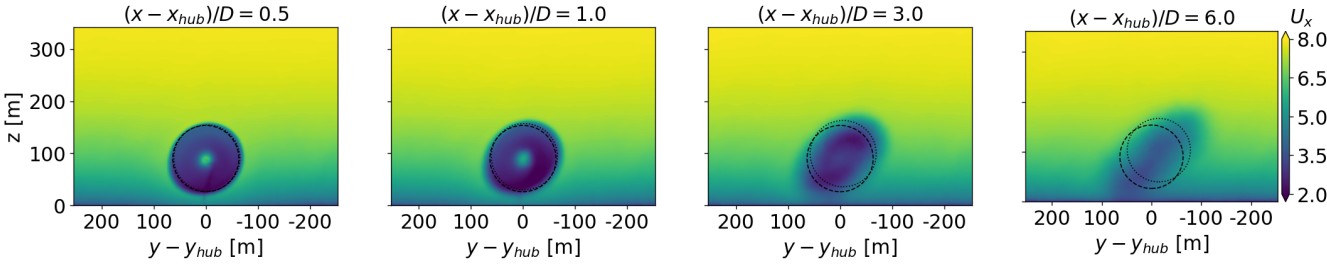

**Figure 21.** Contours of the streamwise velocity, $U_x$, for the baseline case at four different streamwise locations. The dashed line corresponds to the rotor disk centered at the hub height location and the dotted line corresponds to the rotor disk centered around the wake centers.

velocity over the entire extent of the streamwise domain. In fact, by $14D$ downstream, the dominant mode in terms of energy and entrainment for all AWM strategies and the baseline case is $\kappa_\theta = -1$. These results suggest that strategies that excite the $\kappa_\theta = -1$ mode, such as the counter-clockwise helix, may be the most effective in deep-array scenarios.

### 3.3 Power and load performance metrics

Increased turbulent entrainment of mean flow enhances the performance of downstream turbines. This section demonstrates the practical benefits of improved wake recovery using a virtual two-turbine array. The gains in power typically come with the trade-off of increased loads on the upstream turbine due to pitch actuation. Therefore, the effect of different AWM strategies on turbine loads are also discussed.

The results for the second turbine are obtained through a stand-alone OpenFAST simulation of an identical NREL 2.8 MW turbine positioned downstream of the turbine simulated in the LES. AWM is applied solely to the upstream turbine, while the downstream turbine is operated using the baseline controls. Streamwise spacings ranging from $1D$ to $14D$, in increments of $1D$, are analyzed for the two turbine array for each AWM strategy considered on the upstream turbine, as well as for the baseline case, resulting in a total of seventy OpenFAST simulations. At each streamwise location, cross-flow planes are extracted from the LES and used as inflow conditions for the virtual second turbine OpenFAST simulations. This inflow data is collected over a 720s period after the upstream wake has extended beyond $14D$ downstream; the first 120s are discarded as transient data when computing statistics from the outputs of OpenFAST. The lateral location of the second turbine is taken to be aligned with the lateral wake centers (see Figure 8b), placing the second turbine in a nearly fully-waked environment where AWM is anticipated to be most effective (Taschner et al., 2024).

Power increases over baseline operations for the combined two turbine array are shown in Figure 23 and generally follow the trends in rotor-averaged velocity and turbulent entrainment reported in Figure 17. It is important to note that the analyses in the previous sections were aligned with both the vertical and lateral wake centers, whereas the second turbine is only aligned with the lateral wake center, and therefore an exact agreement between the results is not anticipated. For turbine spacings between $1D$ and $9D$, the largest increase over the baseline occurs for the pulse case, with a maximum increase of $3.1\%$ occurring at a turbine spacing of $6D$ (Figure 23). The other AWM strategies lead to increases of around $1\%$ within this region. These results



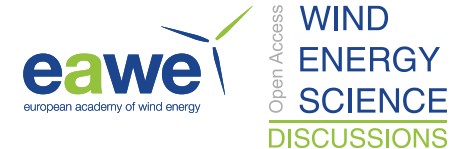

**Figure 22.** Contours of the streamwise velocity, $U_x$, for each AWM case, subtract by the baseline values shown in Figure 21. The dashed line corresponds to the rotor disk centered at the hub height location and the dotted line corresponds to the rotor disk centered around the wake centers.

are consistent with other studies that have examined the pulse method in highly veered ABLs (Brown et al., 2025; Frederik et al., 2025). However, for turbine spacings greater than $9D$, the pulse method exhibits the worst performance amongst all AWM strategies, even underperforming baseline operations. The Helix method is the only AWM case whose total power over the baseline increases as the turbine spacing increases from $1D$ to $12D$, and it is the only method that maintains an increase over the baseline at $14D$ spacing. This behavior is consistent with the sustained increase in modal energy and entrainment for the $\kappa_\theta = -1$ mode observed in the far wake (Figure 19).

To demonstrate the effect of AWM parameters on performance, additional LES of the upstream turbine are performed with pitching amplitudes of $A = 0.5°$ and $A = 2.0°$. Additional stand-alone OpenFAST simulations of a virtual second turbine



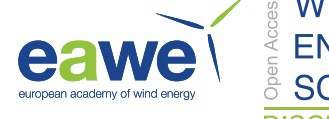

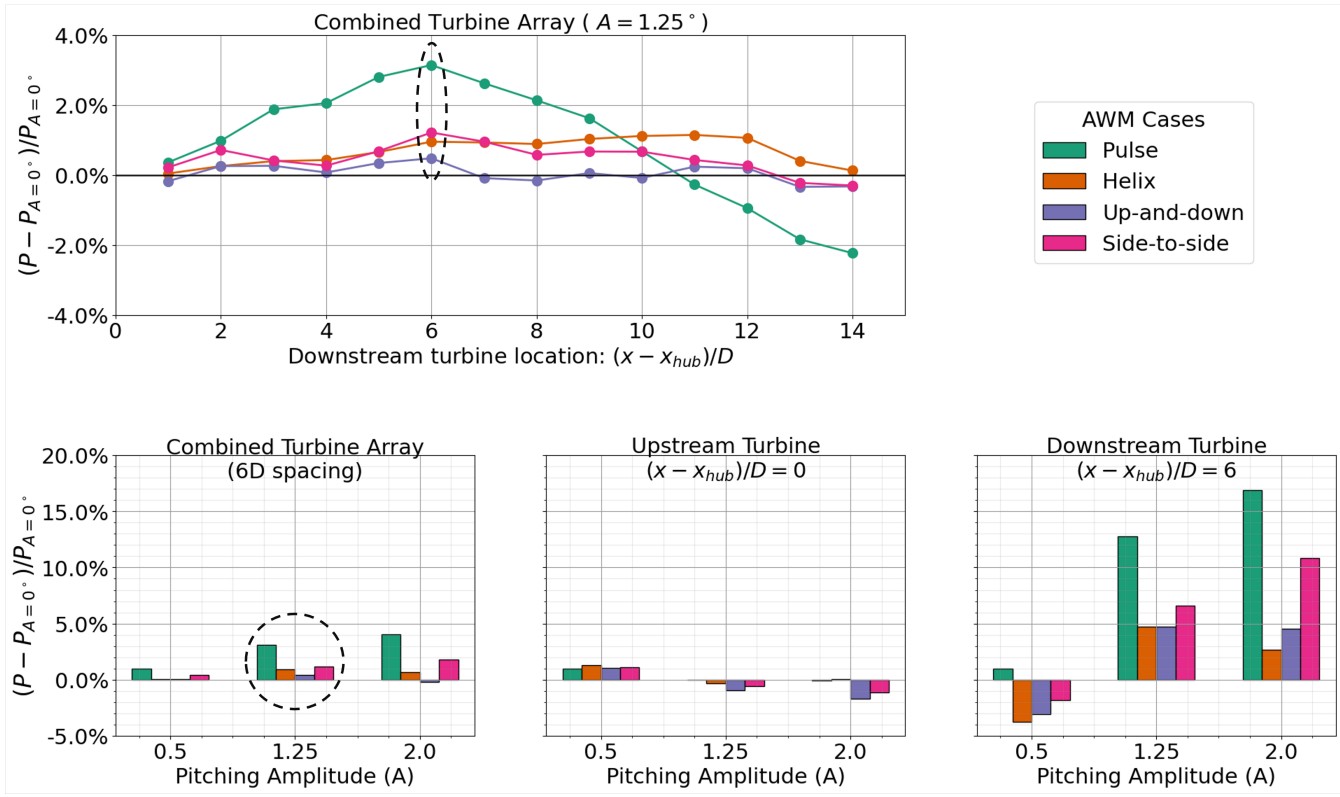

**Figure 23.** (top) Percent change in generated power, $P$, from the baseline case ($A = 0°$) for a two turbine array with the turbine spacings ranging from $1D$ to $14D$. AWM is applied to the upstream turbine with $A = 1.25°$, while the downstream turbine is operated using baseline controls. (bottom) Percent change in generated power from the baseline case for the two turbine array with $6D$ spacing for three different pitching amplitudes applied to the upstream turbine, $A = 0.5°$, $A = 1.25°$, and $A = 2.0°$. Results for the combined two turbine array, the upstream turbine, and the downstream turbine are shown.

located $6D$ downstream are also performed, with the same lateral location as the $A = 1.25°$ case. Power increases over the
baseline operation for the two turbine array range from around $0.5\%$ to $4\%$ depending on the AWM strategy and pitching amplitude (see Figure 23). The pulse method outperforms the other AWM strategies for all pitching amplitudes at this turbine spacing, even though the maximum blade pitch fluctuations for the pulse method is half that of the up-and-down and side-to-die forcings. Again, this agrees with the trends in modal energy and entrainment observed for the pulse method compared to the other AWM strategies. For the larger pitching amplitudes, the $1\%$ to $2\%$ decreases in the upstream turbine performance
are offset by $5\%$ to $17\%$ increases in the downstream turbine (Figure 23). At $A = 2°$, the up-and-down case produces less power than the baseline for the two turbine array. This may be a result of the airfoil operating in a suboptimal region of the lift curve when a large pitching amplitude is applied to the turbine blade in the vertical $0°$ position, where the angle-of-attack is typically near its maximum, especially in the presence of a highly sheared inflow (Taschner et al., 2023). Additionally, note that the variation in the angle-of-attack of the blade and the ABL characteristics around the rotor disk also contributes to the





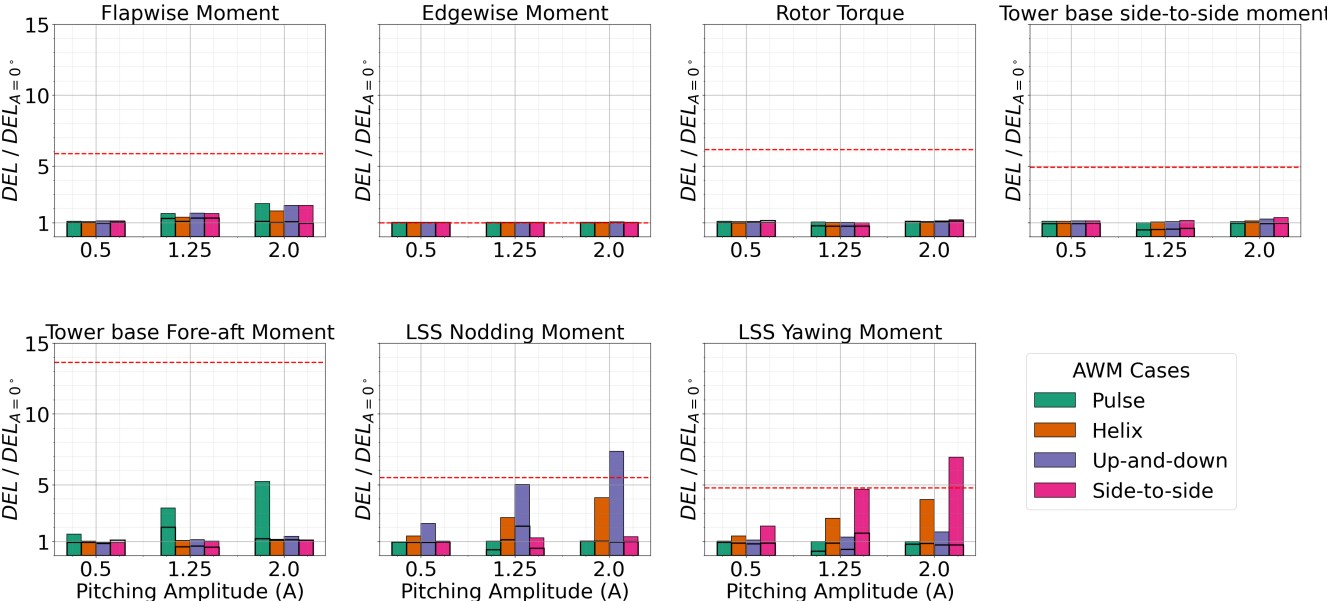

**Figure 24.** Baseline-normalized damage equivalent loads (DEL) for seven different load channels at three different pitching amplitudes, $A = 0.5°$, $A = 1.25°$, and $A = 2.0°$. Solid bars indicate DELs for the upstream turbine, while the DELs for the turbine $6D$ downstream are outlined in black. The red dashed line corresponds to the DELs from a normal turbulence model in a DLC 1.2-like simulation (single seed) with a hub height wind speed of 6.4 m/s, a shear exponent of $0.12$, and a turbulence intensity of $25.90\%$.

difference in axial blade loading and flow response observed between the two mixed-mode forcing strategies. Curiously, the improvements in power at $A = 0.5°$ come from the upstream turbine. This result requires further investigation. However, in terms of wake mitigation, we anticipate that there is a threshold for pitching amplitude, below which the actuation of fluid structures does not yield a reliable net benefit, and a value of $A = 0.5°$ may fall within this regime.

      The increases in power come at the cost of increased loads on the upstream turbine. In Section 2.1, it was shown that the
variations in the blade pitch lead to an increase in the axial force along the blade. However, the dynamic blade pitch fluctuations also affect other load channels on the turbine. In Figure 24, loads are analyzed in terms of the baseline-normalized damage-equivalent loads (DEL) for seven different load channels for each AWM strategy. Flapwise DELs for all AWM cases increase 1.25 to 2.5 times the baseline values as the pitching amplitude increases from $0.5°$ to $2.0°$. The pulse method results in a greater increase in flapwise DEL compared to the helix method at the same pitching amplitudes, which aligns with the increase
in the axial force spectra observed for the pulse case compared to the helix (Figure 6). Large increases in the tower-base fore-aft moment are also observed for the pulse method, whereas the individual pitch control methods lead to increases in the low-speed shaft nodding and yawing moments. However, the DELs for the edgewise moment, rotor torque, and tower-base side-to-side moment do not show as significant increases compared to the baseline. For the downstream turbine, only minor variations in the DELs are observed, likely due to increases in wake TKE, and the changes from the baseline are largely negligible across
all load channels compared to the upstream turbine. In some instances, AWM even reduces the loads on downstream turbines.



Figure 24 also includes the DELs for this turbine using baseline operations in conditions derived from the normal turbulence model in a DLC 1.2-like simulation with the same hub height wind speed as the precursor. This comparison is included to highlight that AWM is particularly well-suited for stable ABL conditions with low TI, where the significant increases in DELs over the baseline largely remain much smaller than the loads on the turbine in typical design environments.

## 4   Conclusions

In this work, an SPOD analysis was developed to track the coherent flow structures induced by AWM throughout the streamwise domain, including their origins in the turbine induction field, growth in the near wake region, and subsequent evolution and energy transfers in the far wake. SPOD is shown to be a particularly useful tool in the context of AWM because it decouples structures in the flow, allowing the wavenumber and frequency inputs to the turbine controller to be translated directly to the wake. The modes directly forced by the blade pitch fluctuations are found to be the dominant flow structures in the near wake region, which originate around $0.5D$ ahead of the turbine. However, as the wake evolves downstream, modal interactions excite other flow structures within the wake, which often exceed the energy of the near wake structures. Further, it is demonstrated that a complete description of AWM effects requires considering correlations between all three velocity components, as significant radial and azimuthal modifications to the wake are induced, in addition to the those in the streamwise direction that align with the axial thrust imparted on the wake.

The SPOD analysis was also connected to conventional wake mixing quantities of interests for different AWM strategies. A new modal decomposition of the radial shear stress flux was developed to measure the contribution of each flow structure in the wake to mean flow turbulent entrainment. This established a quantitative connection between the kinetic energy of coherent structures in the wake and their contribution to turbulent entrainment. The leading flow structures in terms of TKE are found to be primarily responsible for velocity recovery mechanisms within the wake; however, the relationship between modal energy and entrainment is not necessarily directly proportional. In terms of blade loading, a correlation between the spectrum of the axial force along the blade and the spectral characteristics of the induction field was established through the SPOD eigenvalues. Additionally, the modal structure of the induction field suggested that the individual pitch control strategies divert flow around regions of higher blockage, leading to a decrease in the axial blade force compared to collective pitch actuation. A complete end-to-end description of the actuated flow is thus provided. For example, the pulse method (1) induces an axisymmetric modification to the induction field, which (2) results in the largest increase in axial loading amongst AWM strategies; this increase in thrust (3) generates the most dominant coherent structure and largest enhancement of modal energy in the near wake, which (4) leads to greater turbulent entrainment and faster velocity recovery downstream for all AWM strategies; lastly, this (5) ultimately results in the largest increase in generator power for the downstream turbine.

A key observation is that, for the cases considered in this study, the effectiveness of AWM relied on exciting the inherent flow structures in the turbulence of the baseline wake. SPOD, therefore, proved equally useful for analyzing the structure of the wake under baseline operations as it did for the AWM control strategies. For example, the performance of the pulse method was primarily attributed to turbulent entrainment from the $\kappa_\theta = 1$ and 2 modes. These modes were found to be the dominant





ability to enhance the dominant structures in the non-actuated wake. It is important to note that this does not imply the pulse
method will perform optimally under all ABL conditions, rather that the effects of ABL characteristics on the modal structure
of the baseline wake must be considered when designing a control strategy. A broader range of atmospheric conditions should
be examined, including variations in shear, veer, swirl, stratification, TI, and wind speeds to strengthen the relationship between
AWM performance and the coherent structures in the baseline wake. Such insights may enable the optimal design of wind farm
flow control with respect to power and load trade-offs, such as maintaining power benefits with lower pitching amplitudes by
targeting the optimal wake structures.

Similarly, the findings of this study highlight the importance of understanding the nonlinear mechanism responsible for
transferring energy between coherent structures in the wake. While SPOD is used here to track modal TKE and entrainment, it
does not provide an explanation for how energy is transferred between SPOD modes. For example, why does the pulse method
excite the $\kappa_\theta = 1$ and 2 wavenumbers in the flow, even though only the $\kappa_\theta = 0$ wavenumber is directly forced by the controller?
Addressing such questions will be crucial for future work in optimizing AWM strategies, which may rely on understanding
triadic interactions between the forced AWM mode and the other turbulent scales.

Lastly, recent work suggests that a linear stability based reduced order model may be effective for representing the effects
of AWM-induced coherent structures on the mean flow (Cheung et al., 2024c). The SPOD results presented here will be useful
for such model developments, as it will help characterizes the growth rates of coherent structures in the wake, which are inputs
to the linear stability analysis. Similarly, projection-based reduced order models including resolvent or DMD based models
(Gutknecht et al., 2023) would also benefit from the SPOD formulation used in this study. For either approach, the eigenvalue
spectra reported here indicate that it may not be sufficient to only model the forced coherent structures in the wake based $1D$
or $2D$ downstream, particularly for small pitching amplitudes, but that modeling interactions with other large scale structures
in the wake are necessary for accurately representing AWM dynamics.

**Appendix A: SPOD analysis of the individual velocity components**

The SPOD formulation in Section 2.2 includes correlations between all three cylindrical velocity field components, $\boldsymbol{u} = (u_x, u_r, u_\theta)$, in the definition of the cross-spectral density tensor, $\boldsymbol{S}(r, r') = \langle \hat{\boldsymbol{u}}(r, \kappa_\theta, \omega) \hat{\boldsymbol{u}}^*(r', \kappa_\theta, \omega) \rangle$. Here, the SPOD analysis
is repeated for each diagonal element of $\boldsymbol{S}$, namely $S_{11}(r, r') = \langle \hat{u}_x(r, \kappa_\theta, \omega) \hat{u}_x^*(r', \kappa_\theta, \omega) \rangle$, $S_{22}(r, r') = \langle \hat{u}_r(r, \kappa_\theta, \omega) \hat{u}_r^*(r', \kappa_\theta, \omega) \rangle$,
and $S_{33}(r, r') = \langle \hat{u}_\theta(r, \kappa_\theta, \omega) \hat{u}_\theta^*(r', \kappa_\theta, \omega) \rangle$.

The eigenvalues of the cross-spectral density tensor, $\boldsymbol{S}$, and its diagonal components at $St = 0.3$ are shown in Figure A1
across the streamwise extent of the domain. AWM generally induces a consistent modal response in the streamwise, radial, and
azimuthal velocity components throughout the evolution of the wake, although there are differences in kinentic energy between
velocity components. In the near wake region, the dominant flow structure occurs at the forced azimuthal wavenumber in all
three velocity components, i.e., $\kappa_\theta = 0$, $\kappa_\theta = -1$, and $\kappa_\theta = \pm 1$ for the pulse, helix, and mixed-mode actuations, respectively.
The most energetic structures in this region are found in the streamwise velocity component, which is expected since the



azimuthal and temporal variations in the blade pitch are designed to induced a significant axial force along the blade (see Figure 6). However, by $4D$ to $5D$ downstream, the energetic structures (Figure A1) and wake TKE as a whole (Figure 14) are primarily driven by the in plane velocities. For instance, the largest eigenvalue for $S$ across all streamwise locations and

AWM strategies occurs $4D$ downstream for the pulse case at $\kappa_\theta = 1$, which is driven by the radial velocity component, as is the subsequent increase in $\kappa_\theta = 2$ mode for this case (see Figure A1). These two modes are also the dominant structures in $u_r$ for the baseline wake, and were shown to contribute significantly to mean flow turbulent entrainment (see Section 3.2). This suggests that the effectiveness of the pulse strategy is a result of axisymmetric streamwise forcing exciting higher order radial velocity flow structures in the non-actuated wake. Similarly, at $6D$ downstream, $\kappa_\theta = 1$ is the dominant mode for all of the

non-axiysmmetric forcing strategies, which are driven by the radial and azimuthal velocity components. In the far wake, the modal TKE generally returns to the baseline values for all AWM strategies and velocity components.

**Appendix B: Discrete SPOD Solution**

The computational formulation in Section 2.2 results in a discrete complex velocity-Fourier matrix $\hat{\mathbf{U}} \in \mathbb{C}^{N_R \times N_B}$ for each azimuthal wavenumber $\kappa_\theta$ and frequency $\omega$ at each streamwise location (see Towne et al. (2018) for additional details on

the formulation of $\hat{\mathbf{U}}$). Here, $N_R$ is the number of discrete points in the radial domain and $N_B$ is the number of blocks of data resulting from the windowed Fourier transform in time. The analytical eigenvalue problem in equation 4 can then be represented discretely as

$$\mathbf{SW}\boldsymbol{\psi} = \lambda\boldsymbol{\psi}, \tag{B1}$$

where $\boldsymbol{\psi}$ and $\lambda$ are a discrete eigenvector and eigenvalue pair, and $\mathbf{W} \in \mathbb{R}^{N_R \times N_R}$ is positive-definite Hermitian weighting

matrix that accounts for the numerical quadrature of the radial integral. Here, the matrix $\mathbf{W}$ is defined through a one-thirds Simpson's rule. The matrix $\mathbf{S} = \frac{1}{N_B}\hat{\mathbf{U}}\hat{\mathbf{U}}^H \in \mathbb{C}^{N_R \times N_R}$ is a discrete representation of the cross-spectral density tensor, approximated from an ensemble average of velocity correlations over $N_B$ flow realizations. A difficulty in solving equation B1 is that the matrix product $\mathbf{SW}$ is not generally Hermitian. However, a Hermitian problem is achieved by multiplying B1 by $\mathbf{W}^{1/2}$,

$$\tilde{\mathbf{S}}\tilde{\boldsymbol{\psi}} = \lambda\tilde{\boldsymbol{\psi}}, \tag{B2}$$

where $\tilde{\mathbf{S}} = \mathbf{W}^{1/2}\mathbf{S}\mathbf{W}^{1/2}$ and $\tilde{\boldsymbol{\psi}}$. For cases where $N_B < N_R$, there are at most $N_B$ non-zero eigenvalues of $\mathbf{S}$. Therefore, equation B2 can be solved efficiently by performing a low-rank SVD of $\frac{1}{\sqrt{N_B}}\mathbf{W}^{1/2}\hat{\mathbf{U}} = \mathbf{L}\boldsymbol{\Sigma}\mathbf{R}^H$, where $\mathbf{L}$ and $\mathbf{R}$ are the left and right singular vectors and $\boldsymbol{\Sigma}$ is the matrix of singular values. Using this decomposition, $\tilde{\mathbf{S}}$ can be expressed as $\tilde{\mathbf{S}} = (\mathbf{L}\boldsymbol{\Sigma}\mathbf{R}^H)(\mathbf{R}\boldsymbol{\Sigma}\mathbf{L}^H) = \mathbf{L}\boldsymbol{\Sigma}^2\mathbf{L}$. Thus, the eigenvalues, $\lambda$, are given by the square of the singlular values and the eigenvectors, $\tilde{\boldsymbol{\psi}}$, are given by the left singular vectors. Finally, the eigenvectors of the original eigenvalue problem (B1) are obtained by

multiplying $\mathbf{L}$ by $\mathbf{W}^{-1/2}$, i.e., $\boldsymbol{\psi} = \mathbf{W}^{-1/2}\tilde{\boldsymbol{\psi}} = \mathbf{W}^{-1/2}\mathbf{L}$.





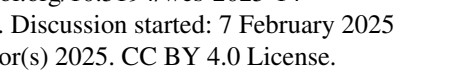

(a): Eigenvalues of **S** at $St = 0.3$ (**u** correlations).

(b): Eigenvalues of $S_{11}$ at $St = 0.3$ ($u_x$ correlations).

(c): Eigenvalues of $S_{22}$ at $St = 0.3$ ($u_r$ correlations).

(d): Eigenvalues of $S_{33}$ at $St = 0.3$ ($u_\theta$ correlations).

**Figure A1.** Eigenvalues of (a) $\boldsymbol{S}$, (b) $S_{11}$, (c) $S_{22}$ and (d) $S_{33}$ corresponding to velocity correlations in $\boldsymbol{u}$, $u_x$, $u_r$, and $u_\theta$, respectively. Eigenvalues at $St = 0.3$ are shown for $\kappa_\theta = 0$ (—), $\kappa_\theta = 1$ (—), $\kappa_\theta = -1$ (—), $\kappa_\theta = 2$ (—), and $\kappa_\theta = -2$ (—),

*Code and data availability.* The code used to perform the LES and SPOD analysis for this study is available as part Exawind software suite: https://github.com/Exawind.



*Author contributions.* Sandia National Laboratories (Gopal Yalla, Kenneth Brown, Lawrence Cheung, Dan Houck, Nathaniel deVelder) conducted the precursor and wind turbine simulations used in this article, developed the SPOD analysis, and wrote the manuscript. Nicholas

Hamilton at the National Renewable Energy Laboratory contributed to gathering and processing the field data that informed the precursor simulation.

*Competing interests.* The authors declare that they have no conflict of interest.

*Acknowledgements.* Sandia National Laboratories is a multimission laboratory managed and operated by National Technology & Engineering Solutions of Sandia, LLC, a wholly owned subsidiary of Honeywell International Inc., for the U.S. Department of Energy's National

Nuclear Security Administration under contract DE-NA0003525.

This work was authored in part by the National Renewable Energy Laboratory, operated by Alliance for Sustainable Energy, LLC, for the U.S. Department of Energy (DOE) under Contract No. DE-AC36-08GO28308.

This research has been supported in part by the Wind Energy Technologies Office within the Office of Energy Efficiency and Renewable Energy. The views expressed in the article do not necessarily represent the views of the U.S. DOE or the U.S. Government. This written

work is authored by an employee of NTESS. The employee, not NTESS, owns the right, title and interest in and to the written work and is responsible for its contents. Any subjective views or opinions that might be expressed in the written work do not necessarily represent the views of the U.S. Government.



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
