# Peer review of "Spectral proper orthogonal decomposition of active wake mixing dynamics in a stable atmospheric boundary layer"

_Wind Energy Science, 2025_

## Author Response (AR1)

April 8, 2025

We wish to thank the reviewer for their helpful and constructive comments. The reviewer's comments and questions are addressed below.

**Reviewer 1**

The manuscript "Spectral Proper Orthogonal Decomposition of Active Wake Mixing Dynamics in a Stable Atmospheric Boundary Layer" by Yalla et al. analyzes four different AWM strategies using SPOD in a stable ABL, relying on numerical simulation results. Apart from the modal analysis, it also investigates the effect of each strategy on the DEL, making an important contribution to further understand and optimize AWM. However, several issues, including one major concern, require attention before publication:

**Major Comment**

1. The right panel of Figure 1 exhibits periodic patterns in the free flow around the wake and streamwise streaks that appear unphysical or at least uncommon. Since the entire study relies on these flow fields, their reliability should be validated. Therefore, the origin of these features needs to be thoroughly investigated, and their potential impact on the results must be assessed. One possible cause could be the grid refinements, whose exact locations should still be clarified in the manuscript. Around the assumed positions of these refinements, small-scale waves can be observed in the flow fields, resembling typical dispersion errors in numerical schemes. To verify the correctness of the flow fields, the authors could compare the current refined cases with a case using a uniform grid throughout the entire domain. However, whether the streamwise streaks are also related to grid refinements remains uncertain and might require further explanation. Flow fields showing a larger area around the turbine and its wake could help clarify this issue.

The spurious oscillations in the flow field that the reviewer noticed were a result of a plotting error in ParaView, and not related to any physical phenomenon or grid refinement issues in the flow. We greatly appreciate the reviewer's attention to detail for noticing this issue. Both the hub-height planes and 3D contour visualizations of the flow have been updated with the corrected fields (see Figure 1 in this document). Additionally, a flow field showing a larger extent of the domain is included here, and overlaid with the regions of refinement for the actuated turbine (Figure 2 in this document). Again, there are no longer nonphysical streaks in the flow field near the refinement boundaries. Information clarifying the refinement regions has been added to the manuscript as well.

The reviewer's larger comment about the effects of transitions between refinement zones resonates well with the authors. In this study, the NREL 2.8MW turbine model has been calibrated for a 1.25m resolution region, and the authors do not have the computational resources to run a uniform-resolution case at this level of refinement. Moreover, the SPOD analysis is primarily focused on large-scale, energy containing structures at low Strouhal numbers, which are expected to be fairly robust to changes in resolution, and the cross-flow planes used for the analysis between 2D and 14D are fully contained in the uniform 2.5m resolution region. The setup used here is also consistent with several other large-eddy simulations performed using the ExaWind suite. Nonetheless, the impact of refinement zones

on resolved turbulence and the requirements for subgrid models warrants a more extensive study, which the authors intend to pursue in future work.

In addition to the update figure, the following text has been added to the manuscript regarding the refinement zones:

• Three distinct levels of isotropic resolution are used to discretize the domain around the turbine: a 5 m background mesh near the domain boundaries; a 2.5 m refinement zone that extends 10D in the streamwise direction and 1.8D in the lateral and vertical directions from the domain center; and a 1.25 m refinement box surrounding the turbine, extending 2D upstream and downstream of the turbine, and 1D in the vertical and lateral directions from the turbine's hub location.

Figure 1: Updated version of Figure 1 in the original manuscript.

Figure 2: Hub-height visualization of the baseline case, overlaid with the refinement regions around the turbine.

**Further Comments**

2. Amplitude Comparison (Lines 142–143 and Figure 5) The "up-and-down" and "side-to-side" cases use double the pitch amplitude compared to the other cases, raising concerns about the fairness of the comparison. This is also critical for the practical implementation of AWM, as the pitch amplitude directly affects the feasibility of applying AWM strategies to real turbines. A stronger initial perturbation likely results in more pronounced wake dynamics while also influencing turbine loads. This discrepancy appears to favor the "up-and-down" and "side-to-side" cases over the helix and pulse strategies. For instance, Figure 12 suggests that the wake differences are more pronounced in these cases. Surprisingly, the "up-and-down" and "side-to-side" cases do not exhibit significantly higher turbine loads, despite the stronger excitation. To ensure a consistent comparison, all cases it would be preferred to maintain the same pitch amplitude, particularly when evaluating turbine performance and structural loads. The authors should further justify their choice of pitch amplitude and, if necessary, provide additional cases with uniform amplitudes for comparison.

The primary motivation for using double the total pitch amplitude in the up-and-down and side-to-side cases is to ensure that each mode in the SPOD analysis is forced with the same amplitudes by the blade-pitch actuations. This allows for a consistent comparison in terms of modal energy and turbulence entrainment between modes and between AWM cases. Since the SPOD results are the main focus of the paper, the authors feel this is the appropriate setup for the study.

However, from a practical standpoint, the reviewer brings up a good point about the fairness of the comparison. To address this, consider the total pitch travel between cases shown in the following figure:

Figure 3: Total pitch travel over one Strouhal period for a single turbine blade

Shown in Figure 3 are the pulse and helix cases at  $A=1.25^{\circ}$ , the side-to-side and up-and-down cases with each mode forced at  $A=1.25^{\circ}$  (2.5° total), and the side-to-side and up-and-down cases with each mode forced at  $A=0.625^{\circ}$  (1.25° total). The side-to-side and up-and-down cases with 2.5° total pitch amplitude exhibit an 18% increase in pitch travel over the helix

case with 1.25° total pitching amplitude. However, the side-to-side and up-and-down cases with 1.25° total pitching amplitude exhibit over a 40% decrease in pitch travel compared to the helix case. Therefore, the authors feel that setting the pitch amplitude for each mode to 1.25° for all AWM cases provides the most consistent comparison in terms of both modal energy and total pitch travel. Information regarding the total pitch travel has been added to the manuscript (see Figure 4 below), and the motivation for the setup of each AWM case has been clarified.

Figure 4: Time series of the blade pitch signal for a single blade for each AWM cases, normalized by the pitching amplitude. One Strouhal period is shown based on the excitation frequency  $\omega_e$ . The black line at  $\Theta/A = 0$  corresponds to the baseline blade pitch signal. The total pitch travel for a single Strouhal period is also shown

In addition to the updated figure, the following text has been added to the manuscript:

- The blade pitch signal for each AWM case is shown in Fig. 4 over a single Strouhal period. For the up-and-down and side-to-side cases, a pitching amplitude of  $A=1.25^{\circ}$  is applied to both the  $\pm 1$  modes. This choice is made to ensure that each mode is forced with the same pitch amplitude, allowing for a consistent comparison of modal energy and entrainment between modes and AWM cases in the SPOD analysis developed in Section 2.2. As a result, the total pitch amplitude for the up-and-down and side-to-side cases is twice that of the pulse and helix cases; however, this corresponds to only an 18% increase in total pitch travel over the helix case (see Fig. 4).
- 3. Several quantities are computed using a rotor disk centered around the wake center instead of the turbine center. Since the wake center also displaces in the vertical direction —something a downstream turbine cannot follow— a turbine-centered approach may provide more applicable insights. This would also help relate the results from Section 3.2 to those in Section 3.3. Would the conclusions change if the rotor disk were centered around the turbine?

This is a good question from the reviewer concerning where the wake analysis in Sections 3.1 and 3.2 should be centered on, and where the downstream turbine in Section 3.3 should be placed. To be clear, all the results in the manuscript are aligned laterally with the wake, rather than the upstream turbine. This includes the positioning of the downstream turbine in the two-turbine array study presented in Section 3.3. Specifically, as the second turbine is moved in the streamwise direction, its lateral placement is adjusted to follow the center of the wake. This is consistent with the SPOD and entrainment analysis discussed in the preceding sections. The decision to align the results with the wake was made because wake control methods are most useful in fully-waked environments. For example, consider the baseline-normalized rotor-averaged velocity for a rotor-disk aligned with the upstream turbine

Figure 5: Baseline normalized rotor-averaged velocity for a rotor-disk aligned with the wake center and a rotor-disk aligned with the turbine. In each case, the rotor-disk is aligned vertically with the turbine hub-height. The results for the pulse and helix mixing strategies are shown.

compared to a rotor-disk aligned with the lateral-wake center for the pulse and helix strategies (see Figure 5 in this document). At all downstream locations, there is a greater improvement over the baseline for the wake-aligned rotor-averaged velocity than the turbine-aligned rotor-averaged velocity. Note that each rotor disk is aligned vertically with the hub-height in this example. The authors have clarified that the turbine-aligned configuration is not the fully-wake environment in this flow, particularly due to high degree of veer, and that the lateral position of the second turbine in Section 3.3 is changed with downstream position.

Furthermore, we would like to note for the reviewer that the primary trends observed in the baseline-normalized eigenvalues from the SPOD analysis remain largely unchanged when aligned with the turbine's hub location, rather than the wake center (see Figure 6 in this document). This is particularly true in the near-wake region where the wake has not significantly displaced from the turbine. While we believe that Figure 6 does not need to be included in the manuscript, we have provided it here for the reviewer's reference to highlight that the

Baseline-normalized eigenvalues of  $\bf S$  at St = 0.3 centered with the turbine hub

Figure 6: Baseline-normalized SPOD eigenvalues centered on the turbine hub location instead of the wake center.

Figure 7: (top) Percent change in generated power, P, from the baseline case  $(A=0^{\circ})$  for a two turbine array with the turbine spacings ranging from 1D to 14D. AWM is applied to the upstream turbine with  $A=1.25^{\circ}$ , while the downstream turbine is operated using baseline controls. (bottom) Percent change in generated power from the baseline case for the two turbine array with 6D spacing for three different pitching amplitudes applied to the upstream turbine,  $A=0.5^{\circ}$ ,  $A=1.25^{\circ}$ , and  $A=2.0^{\circ}$ . Results for the combined two turbine array, the upstream turbine, and the downstream turbine are shown.

results are somewhat robust to this choice.

Lastly, the reviewer is correct that there is a difference in the vertical location that the analysis in Section 3.1 and 3.2 is centered around, and in the placement of the second turbine in Section 3.3, which cannot follow the wake's vertical movements. The authors acknowledge this difference, but feel it is still appropriate to center the wake analysis with the wake's vertical and lateral centers to track the primary flow structures throughout their streamwise evolution, since the SPOD analysis is the primary focus of the manuscript. Notably, the vertical displacement of the wake is much smaller than it's lateral movement, and the performance of the OpenFAST turbine is comparable to the rotor averaged velocity around the wake center. To help strengthen the connection between Sections 3.2 and 3.3, the authors have included the results for the downstream turbines power in the Section 3.3, not just the combined power of the two-turbine array (see Figure 7 in this document). The trends follow very closely with the rotor-averaged velocity reported in Section 3.2.

In addition to the updated figure, the following text is included in the updated manuscript regarding this comment:

• At each streamwise location, cross-flow planes are extracted from the LES and used as inflow conditions for the virtual second turbine OpenFAST simulations. This inflow data is collected from the LES over a 600s period, the first 120s of which are discarded as transient data when computing statistics from the outputs of OpenFAST. All statistics reported here are averaged over seven Strouhal periods. Importantly, the lateral position of the second turbine is adjusted at each streamwise location to correspond with the wake centers identified in Fig. 7b. This decision places the second turbine in the most waked environment at each streamwise location, which is where AWM is expected to be the most beneficial [Taschner et al., 2024]. Recall that, in this flow, the turbine-aligned configuration does not correspond to the most waked conditions for the second turbine, primarily due to the large degree of wind yeer.

The change in power from baseline operations for both the upstream and downstream turbine, as well as for the combined two-turbine array, is shown in Fig. 22. It is important to note that the downstream turbine cannot follow the vertical movements of the wake, so an exact agreement with the wake analysis resulted presented in Sections 3.2 is not anticipated. Nonetheless, the trends in power for the downstream turbine closely align with the rotor-averaged velocity centered with wake (compare Figs. 16b and 22). For turbine spacings between 1D and 9D, the largest increase over the baseline occurs for the pulse case, with a maximum increase of 3.9% occurring at a turbine spacing of 4D. The other AWM strategies lead to increases of around 1% within this region. These results are consistent with other studies that have examined the pulse method in highly veered ABLs [Brown et al., 2025, Frederik et al., 2025]. However, for turbine spacings greater than 10D, the pulse method underperforms baseline operations. The helix method performs the best between 10D and 12D, which is consistent with the sustained increase in modal energy and entrainment for the  $\kappa_{\theta} = -1$  mode observed in the far wake.

- For each streamwise location in the wake, the polar coordinates are defined around the center of the wake, rather than a fixed y-z coordinate at all locations. This approach is taken because the turbulence in the wake, including the AWM-induced flow structures, will follow the mean movements of the wake, and we are interested in tracking their properties downstream. Further, the path the actual wake travels dictates the worst-case position for a downstream turbine (and the best-case scenario for wake control), which further motivates the choice to use the wake position as the centering point for the analyses herein.
- 4. The chosen SPOD parameters resolve Strouhal numbers up to St = 0.15. Could the observed noise in the higher eigenvalues in Figure 20 be related to unresolved frequencies beyond this range? A sensitivity analysis on the SPOD parameters, such as the length of the time series and their overlap, would help determine whether adjustments can mitigate noise and furthermore strengthen the paper's overall contribution.

The authors would like to clarify that the oscillations in Figure 20 of the original manuscript are not a result of noise or lack of convergence, but rather the ordering of modal-indices based on  $\tau_{j,j}$  versus  $|\lambda_j|$ . Consider Figure 8 in this document, which shows the spectra of these two quantities 3D downstream, ordered in two different ways. In the top row of Figure 8, the modes are ordered based on their energy content, represented by the corresponding eigenvalue  $|\lambda_j|$ . In the bottom row of Figure 8, the modes are ordered based on their contribution to entrainment, represented by the quantity  $\tau_{j,j}$ . In each case, it may appear as if there is noise

Figure 8: Energy and entrainment spectra defined by  $|\lambda_j|$  and  $\tau_{j,j}$ , respectively, for the leading 75 eigenvalues at  $(x - x_{hub})/D = 3$ .

in the spectra for the quantity that the indices are not ordered by, i.e.,  $\tau_{j,j}$  in the top row and  $|\lambda_j|$  in the bottom row. However, this is because the SPOD formulation only guarantees a monotonic ordering of modes based on eigenvalues, and there is no reason to expect  $\tau_{j,j}$  to line up with this ordering, especially for smaller turbulent scales. Moreover,  $\tau_{j,j}$ , is not necessarily a positive quantity like  $|\lambda_j|$ , so rapid roll-offs in the spectra occur on a log-scale for any modes that lead to a net turbulent entrainment of mean velocity out of the wake. Throughout the manuscript, the modal index, j, is consistently ordered based on the SPOD eigenvalue, as is most natural, and  $\tau_{j,j}$  is plotted with respect to this ordering.

Additionally the authors would like to clarify that Strouhal numbers up to 19.7 are resolved in this study, which should be more than sufficient for representing the larger indices j that generally correspond to smaller spatio-temporal structures. Moreover, the eigenvalues associated with these scales convergence much more rapidly than the small indices j, so that convergence is generally not an issue in this region of the spectra. The SPOD parameters in this study were chosen to maximize the number of "blocks" over the 1,100s time interval while ensuring good temporal resolution of the forcing Strouhal number, 0.3, and this justification has been added to the manuscript.

The following text has been included in the manuscript to clarify the quantity  $\tau_{jj}$  and the choice of SPOD parameters

• Similarly to  $\mathcal{T}$ , the net mean flow entrainment into the rotor disk from  $\mathcal{T}_{j,j}$  can be quantified by the scalar measure  $\tau_{j,j} = \frac{1}{2\pi} \int_0^{2\pi} \mathcal{T}_{j,j} (r = D/2, \theta) d\theta$ . While the eigenvalue,  $|\lambda_j|$ , is a scalar measure of modal TKE, the scalar quantity  $\tau_{j,j}$  represents the contributions to the net radial shear stress flux from the *j*th SPOD mode. The quantity  $\tau_{j,j}$  is positive for any mode that leads to a net turbulent entrainment of mean velocity into the wake,

and negative for modes leading to the entrainment of mean velocity out of the wake.

- A comparison between modal energy,  $|\lambda_j|$ , and modal entrainment,  $\tau_{jj}$ , across a wider range of wavenumbers and frequencies is shown in Fig. 19. It is generally found that the dominant coherent structures in terms of TKE are also responsible for entraining the most mean flow. However,  $\tau_{j,j}$  does not strictly decrease with the eigenvalue index, j, like  $|\lambda_j|$  does; there are several instances across streamwise locations and AWM cases where the flow structure that entrains the most momentum is not the one with the largest modal energy, although it typically falls within the leading 5-10 eigenvalues. It is important to recall that the SPOD formulation only guarantees an ordering of modes based on energy, not entrainment, so there is no reason to expect, in general, that  $\tau_{j,j}$  decay monotonically with j like  $|\lambda_j|$ . This is particularly evident for the large j-indices shown in Fig. 19. Once the eigenvalue spectrum has decayed by roughly an order of magnitude, SPOD modes even begin to entrain mean flow out of the wake, as indicated by the negative values of  $\tau_{i,j}$ .
- Figure 19: Energy and entrainment spectra defined by  $|\lambda_j|$  and  $\tau_{j,j}$ , respectively, for the leading 75 eigenvalues at  $(x x_{hub})/D = 1$ , 3, and 6. The spectra are normalized on the maximum value of  $|\lambda_j|$  and  $\tau_{j,j}$  for each AWM case and streamwise location. Note that  $\tau_{j,j}$  is not a strictly positive quantity like  $|\lambda_j|$ , so rapid roll-offs in the spectra occur on a log-log scale for any modes contributing to a net entrainment of mean velocity out of the wake.
- ... Each block of data corresponds to 128s of simulation time, with a 64s overlap between consecutive blocks. This configuration allows for the resolution of Strouhal numbers down to St = 0.15 within each block. The SPOD parameters discussed here were primarily chosen to maximize the number of blocks over the 1,100s time interval, while also ensuring good temporal resolution of the forcing Strouhal number, 0.3.
- 5. Figure 23: Power Improvements at A=0.5 Degrees The reported power improvements at the actuated turbine at A=0.5 degrees require further verification. The authors suggest that this might be a regime where the wake does not yet respond to the actuation, yet the downstream turbine loses power, which indicates that the wake reacts, but in an unexpected and unwanted way. Additional analysis, such as energy entrainment or rotor-averaged velocity through the wake, might help clarify this behavior. If verification is not possible, the authors should consider removing the A=0.5 degree case from the manuscript.

Upon further investigation, an error was identified in the computation of the statistics for the two-turbine array, which resulted in an artificial increase in the power of the upstream turbine at  $A=0.5^{\circ}$ . The error involved time-series data being averaged over different intervals between each AWM case and the baseline case, leading to inconsistent comparisons. In the updated results (Figure 7 in this document), the averaging interval has been fixed so that all cases, including the baseline, are averaged over an even number of Strouhal periods. As a result, there is no longer a significant increase in the power of the upstream turbine at  $A=0.5^{\circ}$ . Moreover, the power of the downstream turbine now increases with pitching amplitude for all forcing strategies, as is expected. The authors greatly appreciate the reviewer's suggestion to re-examine these results, leading to these important clarifications.

The text in Section 3.3 has been update to reflect the new results. See the latexdiff document for a complete list of changes.

6. The DEL study does not examine the effects on pitch bearings, which have been identified as a limiting factor in AWM application. Including this additional load channel would enhance the paper's contribution.

The authors have added a pitch travel metric to the comparisons between AWM cases (see Figure 4 in this document), and kept the DEL study limited to the load channels reported from OpenFAST. The authors acknowledge that pitch travel is not equivalent to pitch-bearing usage, but feel it serves as a sufficient metric for quantifying the difference in pitch-wear between the AWM cases for the purposes of this manuscript.

7. Section 3.3 presents the effect of AWM on the DEL. However, the manuscript does not describe how the DEL is computed. Please add this information to the Methodology or the Appendix, or at least include a reference.

The following two references have been added to the manuscript that the authors followed to compute Damage Equivalent Loads

- Freebury, Gregg, and Walter Musial. "Determining equivalent damage loading for full-scale wind turbine blade fatigue tests." 2000 ASME wind energy symposium. 2000.
- Ennis, Brandon L., Jonathan R. White, and Joshua A. Paquette. "Wind turbine blade load characterization under yaw offset at the SWiFT facility." Journal of physics: Conference series. Vol. 1037. No. 5. IOP Publishing, 2018.

These are referenced in the following part of the manuscript:

- In Fig. 18, loads are analyzed in terms of the baseline-normalized damage-equivalent loads (DELs) for seven different load channels for each AWM strategy. DELs are computed following Freebury and Musial [2000] and Ennis et al. [2018]. Flapwise DELs for all AWM cases increase 1.25 to 2.5 times the baseline values as the pitching amplitude increases from 0.5° to 2.0°.
- 8. Consider restructuring the order of the Figures to align more logically with the text's progression.

The authors have restructured Figures 9 and 10 in the original manuscript to align with the reviewer's suggestion. The baseline-normalized eigenvalues are now shown together in one figure, with the top row corresponding to the near-turbine region,  $-1 \leq (x-x_{hub})/D \leq 1$ , and the bottom row corresponding to the entire streamwise domain,  $-5 \leq (x-x_{hub})/D \leq 14$  (see Figure 14 in this document). Likewise, the globally-normalized eigenvalues are also shown together in one figure using the same layout (see Figure 13 in this document). We feel this better aligns the figures with the order the analysis. Moreover, Figure 2 in the original manuscript has been removed altogether, in response to the reviewer's later suggestion. The authors will do their best to position the figures as close to the locations where they are discussed in the paper, and will work with the journal's editorial team to correct the typesetting and placement of the figures in the final version of the manuscript. The text in section 3.1 has also been revised to more closely follow the ordering of figures.

**Minor Issues and Clarity Improvements**

The authors have done their best to address all minor issues raised by the reviewer. These suggestions were helpful in improving the clarity of the manuscript. Further responses to a selection of the reviewer comments are included below.

• Line 34: The term "turbine layout" likely refers to "wind farm layout."

This sentence in the text now reads:

These results vary significantly with wind farm layout, turbine model, and ABL condition.

• Line 35: The statement about AWM having a "Additionally, the design space for AWM is considerably larger than that of other WFFC strategies. Common implementations rely on at least four relevant design parameters to control..." is unclear. Further elaboration is needed.

This section now reads,

These results vary significantly with wind farm layout, turbine model, and ABL condition. Additionally, the design space for AWM is considerably large. While wake steering, for example, depends on setting the yaw angle, common implementations of AWM involve several relevant design parameters to control the blade pitch fluctuations, such as the pitch amplitude, forcing frequency and wavenumber, clocking angle, and waveform of the pitch signal (see Cheung et al. [2024] and Section 2.2. Given the complex interactions between wind conditions, turbine layout, and turbine control parameters, we therefore cannot expect to rely solely on parametric studies to optimize AWM. Instead, a deeper understanding of the physical mechanisms behind the power and load trade-offs of AWM is necessary to effectively navigate the design space.

• Line 207: Planes are sampled at a 1.25m resolution, while the grid resolution is at least in parts coarser (2.5m or more). Is it correct that the planes are sampled at a higher resolution than the grid, and if yes, why?

This has been corrected, thank you. The planes are sampled at a spatial resolution consistent with the local grid resolution.

This sentence now reads:

Given the computational setup in Section 2, SPOD is performed on time series of cross-flow planes extracted from the LES at multiple streamwise locations ranging from  $-5 \le (x-x_{hub})/D \le 14$  (see Fig. 6). These y-z planes are sampled at a spatial resolution consistent with the local grid resolution and a frequency of 2Hz over the 1,100s of simulation time...

• Line 212: "Each" should be lowercase ("each").

This has been correct in the manuscript.

• Lines 470: The term "DLC 1.2-like simulations" is not explained. Further clarification is necessary. The following paragraph has been added to this section:

To provide context for these loads, the DELs for this turbine in a DLC 1.2 environment are also included in Fig. 18. A DLC 1.2 simulation is typically used to evaluate the structural integrity and performance of the wind turbine under expected operational conditions, ensuring that it can withstand the loads and stresses it will encounter during its operational life. The

Figure 9: Baseline-normalized damage equivalent loads (DEL) for seven different load channels at three different pitching amplitudes,  $A = 0.5^{\circ}$ ,  $A = 1.25^{\circ}$ , and  $A = 2.0^{\circ}$ . Solid bars indicate DELs for the upstream turbine, while the DELs for the turbine 6D downstream are outlined in black. The red dashed line corresponds to the baseline-normalized DELs from a normal turbulence model in a DLC 1.2-like environment (single seed) with a hub height wind speed of 6.4 m/s, a shear exponent of 0.12, and a turbulence intensity of 25.90%.

DLC 1.2 results are computed here with a single seed for the turbine using baseline controls in conditions derived from the Normal Turbulence Model with a hub height wind speed of 6.4 m/s and a turbulence intensity of 25.90%. This comparison is included to highlight that AWM is particularly well-suited for stable ABL conditions with low TI, where the significant increases in DELs over the baseline largely remain much smaller than the loads on the turbine in typical design environments.

- Figure 6: What causes the fluctuations in modal blade loads around x = 0.6 r/R, which are absent in the baseline but appear in all AWM cases?
  - This behavior has been previously observed for the NREL 2.8MW turbine model, even for canonical flow conditions [Cheung et al., 2024], and is currently under further investigation. The authors suspect that these fluctuations may be an artifact of the blade model rather than a result of the computation of the axial force spectra. The authors are interested to see if these fluctuations arise for other turbine models as well.
- Figure 10: The Greek "kappa" symbol is not properly compiled in the caption. This has been corrected in the manuscript.
- Figure 24: The upper limit of the y-axis is significantly higher than the highest bar, creating excessive white space and making it difficult to distinguish between the bars for the upstream and downstream turbines. Adjust the upper limit for clearer presentation of the results.
  - Figure 24 of the original manuscript has been revised so that the y-axes are no longer unified across load channels (see Figure 9 in this document). Please note that some of the additional

white space in the figures arises from the values of the DLC 1.2 case, which provides important context for the baseline loads.

- Reevaluate whether all figures contribute meaningfully to the study (e.g., Figure 2, left panel). Figure 2 has been removed from the manuscript.
- Consider citing the work by Muscari et al. "Physics-Informed DMD for Periodic Dynamic Induction Control of Wind Farms" (DOI: 10.1088/1742-6596/2265/2/022057) in the Introduction. Citation: https://doi.org/10.5194/wes-2025-14-RC1.

Thank you for the relevant reference — a citation has been added to the manuscript.

Additionally, the following text has been added to the introduction:

In the context of AWM, Cheung et al. [2024] demonstrated in a canonical flow that SPOD could be used to track the modal growth of instabilities in the wake. Other related data-driven methods also exist, such as space-only POD and Dynamic Mode Decomposition (DMD), which provide spatial and temporal orthogonalizations of the flow, respectively. DMD has recently been used to analyze flow structures [Muscari et al., 2022] and model wake dynamics [Gutknecht et al., 2023] within the context of AWM. While these methods are effective at identifying either spatially coherent structures or dynamically meaningful temporal structures in the flow, the SPOD identifies structures that evolve coherently in both space and time. Furthermore, as demonstrated by Towne et al. [2018], the SPOD structures correspond to an ensemble-average of DMD structures, so that the SPOD structures are both dynamically significant, unlike those identified by space-only POD, and provide an optimal description of the energy content in the flow.

**Reviewer 2**

The manuscript by Yalla et al analyzes different AWM strategies by using SPOD to isolate different modes in the flow at different frequencies related to the AWM excitation frequency. The work has significant scientific value as it furthers our understanding of the flow dynamics associated with AWM. On the other hand, the current manuscript leaves some room for improvement in terms of structure, and some of the results presented require additional analysis or explanation. In short, I feel like this manuscript is a very rough diamond. It should in my opinion be published after the authors perform some serious polishing. I voted to accept subject to minor revisions, because I believe in the value of this manuscript, but please note that some of the changes I feel are necessary are leaning towards "major revisions".

**Major Comments**

You mention in the second line of the introduction that power losses are particularly problematic
in stable ABL, most commonly found in offshore wind farms. Yet, in section 2.1, you derive the
simulations from measurements from an onshore farm. Please explain or justify why you did not
choose to run simulations to match offshore conditions, where AWM would be expected to be
most effective.

Wake mixing strategies are most beneficial in stable ABLs with low TI, where wakes are known to persist far downstream of turbines. While stable ABLs are indeed common offshore, they also occur in onshore environments. Consider, for example, the histograms in Figure 10, which show wind speed and turbulence intensity measurements taken offshore at the NY Bight and onshore measurements taken as part of the AWAKEN campaign. Both cases exhibit low TI, region 2 wind speeds that are representative of stable ABL conditions and are conducive to wake mixing strategies.

The decision to utilize measurements from the AWAKEN domain was, in part, driven by funding logistics and project scope; however, the authors are confident that the AWAKEN measurement still lead to valid stable conditions, especially for the 1,100-seconds needed for the simulations in this study.

To avoid any potential confusion regarding the context of our simulations, we have revised the introduction to remove the specific mention of offshore conditions.

2. Similarly, the choice of wind speeds, wind directions, TI's, and veer that is studied needs more justification. I can see how these conditions would correspond to a stable ABL, but I don't see how these specific conditions are necessarily prerequisites for a stable ABL. Is there literature available describing ABL's in the AWAKEN experiment that supports this choice? Or are there other reasons you have chosen these conditions specifically? Also, please define the definition of wind direction and why these directions are more likely to result in stable ABL's than other WD's. Furthermore, please clarify whether the 230 minute dataset used is continuous or a combination of different subsets. Finally, in line 102, you mention that the resulting veer is 9 degrees, which is substantially lower than the threshold you mention earlier. Please explain why this is the case and how this still accurately represents the dataset, as we have seen in recent publications by Brown et al (2025) and Frederik et al (2025) that lower veer can have a large impact on how well different AWM strategies work.

Figure 10: Comparison of wind speed and TI measurements taken at onshore as part of the AWAKEN campaign and offshore at the NY Bight

The authors agree that insufficient information was provided in the original manuscript regarding the target wind conditions and their derivation from a field campaign. The text has been rewritten to improve clarity and accuracy. We believe the reviewer's questions are answered directly by the new text below:

The streamwise and lateral boundary conditions are defined by an inflow condition extracted from an initial precursor simulation. The target conditions for the precursor simulation are derived from stable ABL measurements taken in 2021 concurrent with the American Wake Experiment (AWAKEN) campaign [Moriarty et al., 2024]. Specifically, the atmosphere was sampled at the Department of Energy's Atmospheric Radiation Measurement (ARM) Southern Great Plains (SGP) C1 site [Facility, 2024] using a Doppler profiling lidar and CO2 flux stations. A number of filters were applied to isolate 30-minute bins corresponding to high-quality measurements and periods of likely interest for wake control. Balancing the need for appreciable sample size with that for retaining the realism of a specific, observable ABL condition, the latter filters isolated a wind condition with a frequency of occurrence: wind speed between 6 and 6.7 m/s, wind direction ranging from 100° to 260° (assuming a clockwise definition of wind direction with 0° corresponding to northerly flow), and turbulence intensity (TI) from 0% to 7%. The 1200 minutes of (non-contiguous) data that meet the above criteria are used to establish target statistics for the precursor simulation, and these are a hub-height

wind speed of 6.4 m/s, a TI of 4.4%, a shear exponent of 0.19, and 17° of veer over the rotor disk. The low TI and high veer in this target are congruent with the southerly, stable ABL conditions typically observed at the SGP site [Krishnamurthy et al., 2021], and these conditions offer a strong opportunity to demonstrate a use-case for wake-control technology.

The surface roughness height (0.0015 m) and cooling rate (4.16  $\times$  10-5 K/s) parameters are calibrated in the precursor simulation of the ABL to ensure good agreement between the simulated and measured flow statistics; this calibration results in inflow data with an average hub-height wind speed of  $U_{\infty}=6.4 \text{m/s}$ , a TI of 3.5%, a shear exponent of 0.17, and 9° of veer over the rotor disk. Notably, the simulated ABL exhibits less veer than the measurement data, and this is due to the numerical forcing scheme used to drive the ABL towards the target statistics (primarily TI and shear), which limits the amount of simulated veer that can be achieved while maintaining a realistic ABL velocity profile; such difficulty achieving high veer was also encountered in Brown et al. [2025] and Frederik et al. [2025]. Alternate ABL forcing schemes, such as those which incorporate meso-scale information in a direct assimilation approach, may be able to match the veer and wind speed profiles, but are not studied here. In addition to the above summary statistics, a good agreement between the vertical profiles of the simulated and measured ABL is found (see Fig. 2).

- 3. I agree with Reviewer 1 that not all figures add value to the paper. Consider removing Figure 2, 4 (unless you add the Cp/Ct curves of the turbines in the actual WF for comparison), 7, 21, and 22. Furthermore, I would be interested to also see a comparison between data and simulation in the time and/or frequency domain in Figure 3, not just a comparison of averages.
  - The authors have removed Figure 2 from the original manuscript and thank the reviewer for this suggestion. The other figures have been retained. See below for additional comments regarding figures 7, 21, and 22. In the current study, the most meaningful comparisons between the simulated and measured ABL data would be of the 10-min averaged statistics, both in terms of the vertical velocity profiles and the hub-height quantities such as shear, turbulence intensity, and wind speed. Because the precursor development used constant surface roughness, surface temperature properties, and target wind speed/direction during the spin-up phase, the time evolution of the ABL leading up to the simulation period is not expected to match the same behavior as the measured data. In other ABL forcing schemes, such as approaches which use meso/micro-scale coupling, a direct comparison of the ABL properties is more meaningful. Such forcing schemes are under active development and are planned for use in future work. We have clarified the information regarding the measurement data and the formulation of the precursor in the manuscript (see the text above).
- 4. I have concerns about the azimuth angle approximation stated on line 151-153. I have seen simulations in similar wind conditions, and especially if you are implementing AWM, I would expect non-negligible changes in rotor speed. Even tiny changes could affect the outcome of your Fourier approximation as you are in practice taking the transformation of a mode/azimuthal wave frequency that is variable over time. I would like to see what effect these variations have on the Fourier transformation, i.e., how well of an approximation Eq.(3) is w.r.t. Eq.(2). In fact, my guess would be that most of the non-periodicity that you describe in line 156 is caused by this fluctuation in rotor speed, not by the non-uniform inflow. Have you checked the periodicity when you don't use the rotor speed approximation? How wide is your window now? I believe the rotor speed approximation might be reasonable to make in combination with a windowed transform,

but would like to see this better studied or more accurately described in this section.

The authors thank the reviewer for their suggestion to re-examine the formulation of the blade-loading spectra. Although the mean-rotor speed approximation provides a convenient way of translating timeseries data from OpenFAST to spectra using a single Fourier-transform, it is not strictly necessary. Moreover, it may not be exact due to fluctuations in the rotor speed, as the reviewer points out. To address this, the authors have developed an alternative method for the computing the spectra by forming  $F_x(r, \theta, t)$  directly using the instantaneous rotor speed signal to inform the azimuthal position of the turbine blade. Consequently, the spectra of the axial blade force at a particular azimuthal wavenumber,  $\kappa_{\theta}$ , and frequency,  $\omega$ , can be computed by performing a Fourier-transform in both the azimuthal and temporal directions, i.e.,

$$\hat{F}_x(r,\kappa_\theta,\omega) = \int \int F_x(r,\theta,t)e^{-i(\omega t + \kappa_\theta \theta)}d\theta dt.. \tag{1}$$

The updated results using this formulation are shown in Figure 11 of this document and have been included in the manuscript. No major differences with the original formulation are observed, and the main results and conclusions of this section are unchanged.

Figure 11: Variations in the axial blade loading for each AWM strategy. The individual curves correspond to the magnitude of the Fourier coefficients of the axial force,  $|\hat{F}_x|$ , at St=0.3 for different azimuthal wavenumbers,  $\kappa_{\theta}$ , normalized by the mean axial blade loading,  $|\overline{F}_x|$ .

In addition to the update figure, the following text has been included in the manuscript:

• The blade pitch fluctuations are designed to induce azimuthal and temporal variations in the blade loading at the wavenumbers and frequencies input to the controller. These variations can be examined through the spectrum of the axial force along the blade span [Cheung et al., 2024]. The Fourier representation of the axial force,  $F_x$ , at a particular

radial location, r, blade azimuthal angle,  $\theta$ , and time, t, for a given blade is

$$\hat{F}_x(r,\kappa_\theta,\omega) = \int \int F_x(r,\theta,t)e^{-i(\omega t + \kappa_\theta \theta)}d\theta dt.$$
 (2)

Given a discrete time series of the axial force at each nodal location along the blade,  $F_x(r,t)$ , the blade azimuthal angle is determined using the instantaneous rotor speed signal,  $\Omega$ , as  $\theta(t) = \theta(0) + \int_0^t \Omega(t')dt'$ . Then the Fourier coefficients,  $\hat{F}_x$ , are readily determined by performing a Fourier transform in the time and azimuthal directions.

5. In Section 3.1, I would be very interested to see an analysis of the eigenvalues of the baseline case for different values of St. The main hypothesis of why St=0.3 is optimal, is that it excites natural modes in the flow. Therefore, I would expect that taking the Fourier transform at different frequencies would result in lower values. Adding an analysis of the magnitude of different wavenumbers as a function of Strouhal numbers (say St=0.1 to 0.6) could reinforce or disprove this hypothesis and thus add significant value to the scientific contribution of this paper.

The authors appreciate the reviewer's interest in the eigenvalues of different Strouhal numbers for the baseline case. These results are shown in Figure 12 of this document. It is generally observed that the eigenvalues at St = 0.3 are larger than the other Strouhal numbers, although there is some variability depending on the azimuthal wavenumber and streamwise location.

The authors would like to clarify that the manuscript does not claim that St=0.3 is necessarily the optimal choice, but rather highlights that the performance of AWM strategies when forced at St=0.3 corresponds to their ability to excite natural modes in the flow at this Strouhal number. The value of St=0.3 was chosen based on recommendations in the literature, where it is generally found that Strouhal numbers between 0.2 and 0.35 perform well. However, the optimal value is still an open question. The SPOD parameters in this study were chosen to maximize the number of "blocks" over the 1,100s time interval while ensuring good temporal resolution of the forcing Strouhal number St=0.3. As such, there is insufficient resolution to do a refined SPOD study of the baseline wake for different Strouhal numbers between 0.2 and 0.35. Since the eigenvalues of the baseline at St=0.3 are extensively shown throughout the manuscript, the authors have decided not to include Figure 12 of this document in the final version. However, this type of analysis will be considered for future studies that consider the optimal choice of forcing frequency.

Figure 12: SPOD eigenvalues for the baseline wake at five different Strouhal numbers

6. I suggest restructuring Section 3.1 and its figures, as it is very hard to follow now. I keep having to scroll through pages to get to the figures that are covered in the text. For example, perhaps

you should redo figures 9 and 10 so one of them shows the global EVs for both ranges, and the other the baseline-normalized EVs, as this is also how it's discussed in the text. Similarly, perhaps put fig 11a and 12 together for better comparison. All the other panels of fig 11 don't seem to be discussed, but could probably do with some more explanation as these results are probably interesting but not self-explanatory. In general, I would restructure this section so the analysis is grouped in the same order as the figures. That way, the reader no longer needs to scroll back and forth between pages every couple of lines. I would also consider splitting this subsection into multiple (sub)subsections to better separate different points that you are trying to make.

The authors appreciate the reviewer's constructive feedback regarding the structure of Section 3.1 and the associated figures. We have restructured Figures 9 and 10 in the original manuscript to align with the reviewer's suggestion. The baseline-normalized eigenvalues are now shown together in one figure, with the top row corresponding to the near-turbine region,  $-1 \le (x - x_{hub})/D \le 1$ , and the bottom row corresponding to the entire streamwise domain,  $-5 \le (x - x_{hub})/D \le 14$  (see Figure 14 in this document). Likewise, the globally-normalized eigenvalues are also shown together in one figure using the same layout (see Figure 13 in this document). We feel this better aligns the figures with the order the analysis. The text in section 3.1 has also been revised to more closely follow the ordering of figures.

Regarding the reviewer's second comment, the authors have decided to keep Figures 11a-d in the original manuscript together to illustrate the progression of the dominant SPOD modes through the wake, and have added additional clarifications for panels b-d of this figure. The phase-averaged, baseline-subtracted velocity fields are included on the page immediately following the figure of SPOD modes to hopefully minimize page turning by the reader for this comparison. Furthermore, the authors will do their best to position the figures as close to the locations where they are discussed in the paper, and will work with the journal's editorial team to correct the typesetting and placement of figures in the final version of the manuscript.

7. I question the fidelity of the results presented in Figure 23 and surrounding text. The different pitch amplitude cases raise more questions to me than they answer. First, the upstream power gain at low amplitudes is very peculiar and should be studied. Same goes for the fact that helix and up-down lose downstream power when the amplitude increases. These results make me question the fidelity of the standalone OpenFAST model approach used, as the results do not align with similar studies performed using higher fidelity tools. I think you need to choose to either remove this analysis from the paper, or dive in deeper to explain why these results are different from literature. I recommend doing the former, as this analysis does not align with the main findings of the paper to begin with. Same can be said about Figure 24.

Upon further investigation, an error was identified in the computation of the statistics for the two-turbine array. This led to the two errors pointed out by the reviewer: an artificial increase in the power of the upstream turbine at  $A=0.5^{\circ}$  and a decrease in the downstream turbine's power with pitch amplitude for certain cases. In the original results, time-series data was averaged over different time intervals between each AWM case and the baseline case, leading to inconsistent comparisons of power. In the updated results (Figure 15), the averaging interval has been fixed so that all cases, including the baseline, are averaged over an even number of Strouhal periods. As a result, there is no longer a significant increase in the power of the upstream turbine at  $A=0.5^{\circ}$ . Moreover, the power of the downstream turbine now increases with pitching amplitude for all forcing strategies, as the reviewer expected to see.

(a): Globally-normalized eigenvalues of **S** at St = 0.3 for  $-1 \le (x - x_{hub})/D \le 1$

(b): Globally-normalized eigenvalues of **S** at St = 0.3 for  $-5 \le (x - x_{hub})/D \le 14$

Figure 13: Eigenvalues of **S** at St = 0.3 for the azimuthal wavenumbers  $wkappa_{\theta} = 0$  (—),  $\kappa_{\theta} = 1$  (—),  $\kappa_{\theta} = -1$  (—),  $\kappa_{\theta} = 2$  (—), and  $\kappa_{\theta} = -2$  (—). Each eigenvalue is normalized by the global maximum eigenvalue across all AWM cases and streamwise locations.

(a): Baseline-normalized eigenvalues of **S** at St = 0.3 for  $-1 \le (x - x_{hub})/D \le 1$

(b): Baseline-normalized eigenvalues of **S** at St = 0.3 for  $-5 \le (x - x_{hub})/D \le 14$

Figure 14: Eigenvalues of **S** at St = 0.3 for the azimuthal wavenumbers  $wkappa_{\theta} = 0$  (—),  $\kappa_{\theta} = 1$  (—),  $\kappa_{\theta} = -1$  (—),  $\kappa_{\theta} = 2$  (—), and  $\kappa_{\theta} = -2$  (—). The eigenvalues for each wavenumber and streamwise location are normalized by the corresponding eigenvalues in the baseline case.

To strengthen the connection between these results and those in the previous section we have also included the results for the downstream turbine's power as a function of streamwise spacing, rather than solely presenting the combined power of the two-turbine array. These

Figure 15: (top) Percent change in generated power, P, from the baseline case  $(A=0^{\circ})$  for a two turbine array with the turbine spacings ranging from 1D to 14D. AWM is applied to the upstream turbine with  $A=1.25^{\circ}$ , while the downstream turbine is operated using baseline controls. (bottom) Percent change in generated power from the baseline case for the two turbine array with 6D spacing for three different pitching amplitudes applied to the upstream turbine,  $A=0.5^{\circ}$ ,  $A=1.25^{\circ}$ , and  $A=2.0^{\circ}$ . Results for the combined two turbine array, the upstream turbine, and the downstream turbine are shown.

trends align well with the rotor-averaged velocities presented earlier in the manuscript. See the responses to the minor comments below for a further discussion of the setup used to generate these results.

We greatly appreciate the reviewer's suggestion to re-examine these results, which has led to these important clarifications.

The text in Section 3.3 has been update to reflect the new results. See the latexdiff document for a complete list of changes.

**Minor Comments**

The authors have done their best to address all minor issues raised by the reviewer. These suggestions were helpful in improving the clarity of the manuscript. Further responses to a selection of the reviewer comments are included below.

• Line 54: "phenomenon", not "phenomena"

This has been corrected in the manuscript.

• Line 56-57: I agree with the first reviewer that Muscari et al, 2022 and/or 2025, and/or Gutknecht et al, 2023 should be cited and discussed here, or in line 61-62. It seems to me that these studies use a slightly different method to achieve a similar goal. It would therefore be worthwhile to discuss the differences between and/or advantages and disadvantages of both methods.

Thank you for the relevant references — citations have been added to the manuscript.

Additionally, the following text has been added to the introduction:

In the context of AWM, Cheung et al. [2024] demonstrated in a canonical flow that SPOD could be used to track the modal growth of instabilities in the wake. Other related data-driven methods also exist, such as space-only POD and Dynamic Mode Decomposition (DMD), which provide spatial and temporal orthogonalizations of the flow, respectively. DMD has recently been used to analyze flow structures [Muscari et al., 2022] and model wake dynamics [Gutknecht et al., 2023] within the context of AWM. While these methods are effective at identifying either spatially coherent structures or dynamically meaningful temporal structures in the flow, the SPOD identifies structures that evolve coherently in both space and time. Furthermore, as demonstrated by Towne et al. [2018], the SPOD structures correspond to an ensemble-average of DMD structures, so that the SPOD structures are both dynamically significant, unlike those identified by space-only POD, and provide an optimal description of the energy content in the flow.

- Line 71: Add comma after "2"
   A comma has been added.
- Figure 1: Please include labels on the axes. Consider normalizing by the rotor diameter. Labels have been added to the axes.
- Line 106: change gradient to 7.5 · 10-4
   This change has been made in the manuscript.
- Line 117: Please define all parameters.
   Definitions have been added for the ALM and FLLC parameters.
- Line 123: This equation is not self-explanatory. Although I can see the value of this definition in light of the SPOD modes later on, I feel like this definition of the blade pitching is far less intuitive than the one used in publications by other groups, that uses the MBC/Coleman transformation. Consider relating this equation to that definition, or at least adding blade number subscripts and explaining what the definition of the clocking angle is.

The authors have retained the normal-mode formulation of the blade-pitch because it aligns directly with the azimuthal and temporal frequencies that are analyzed through the SPOD analysis. This justification for using the normal-mode representation over the Coleman-transform representation of the blade pitch has been added to the manuscript. Moreover, the discussion surrounding the clocking angle has been updated and it is clarified that the blade-pitch formulation is applied to each blade.

The following text as been added to the manuscript:

To implement different AWM control strategies on the turbine, a dynamic blade pitch,  $\Theta$ , is specified on top of the baseline pitch set point,  $\Theta_0$ . The dynamic blade pitch can be specified

using either a normal-mode representation [Cheung et al., 2024] or a Coleman-transform representation [Frederik et al., 2020]. Both representations have been implemented in NREL's reference open-source controller (ROSCO v2.8.0; National Renewable Energy Laboratory [2024]), which is coupled to OpenFAST in the LES. In this work, the normal-mode representation is used because the wavenumber and frequency input to the controller align directly with the parameters of the SPOD analysis formulated in Section 2.2. Specifically, the dynamic blade pitch for each blade is specified as

- Line 126: Double use of the word "structure"

  This has been corrected in the manuscript.
- Line 128: Consider changing to  $U_{\infty}$ .

  The symbol  $U_{\inf}$  has been replace with  $U_{\infty}$  throughout the manuscript.
- Line 140, Table 1: I do not understand how the clocking angle creates a non-uniform thrust force across the rotor disk, as it is constant. If I understand correctly, the clocking angle is only relevant for the side-to-side and up-and-down strategies, in which case it is used to have the two modes negate each other in horizontal and vertical direction, respectively. For the other strategies, it only changes the phase of the excitation, which is why I do not understand why it is defined at 90 degrees. For the helix and pulse, the pitch angle can equivalently be written as (when  $\phi_{clock} = 90$  degrees):

$$\Theta(t) = \Theta_0(t) - A\sin(\omega_e t - \kappa_\theta \psi(t))$$

Please clean up this definition.

The reviewer is correct in that, for this study, the clocking angle,  $\phi_{\rm clock}$ , is mostly important for differentiating the side-to-side ( $\phi_{\rm clock} = 90^{\circ}$ ) and up-and-down ( $\phi_{\rm clock} = 0^{\circ}$ ) strategies. For the pulse and helix cases, the clocking angle is only relevant for the initial set-point of the blade pitch, and it is set to 90°. In situations with two-turbines, the clocking angle becomes important for the helix case (and other non-axisymmetric forcing strategies) when "synchronizing" wake control methods between upstream and downstream turbines [van Vondelen et al., 2025]; however, this is not the case here. The role of the clocking angle has been clarified in the manuscript.

The following text has been added to the manuscript:

Four AWM strategies are considered in addition to a baseline case, including a pulse ( $\kappa_{\theta} = 0$ ), helix ( $\kappa_{\theta} = -1$ ), side-to-side ( $\kappa_{\theta} = \pm 1$ ,  $\phi_{\text{clock}} = 90^{\circ}$ ), and up-and-down ( $\kappa_{\theta} = \pm 1$ ,  $\phi_{\text{clock}} = 0^{\circ}$ ) actuation. Here, positive and negative azimuthal wavenumbers denote flow structures that rotate in the same and opposite direction of the turbine, respectively (i.e., clockwise and counter-clockwise when looking downstream). The counter-clockwise helix method is used here, which has been found to consistently outperform its clockwise counterpart [Coquelet et al., 2024]. The pulse forcing is axisymmetric and achieved through collective pitching of the three turbine blades, while the other AWM strategies rely on individual pitch control to create a non-uniform thrust force around the rotor disk. In this study, the clocking angle is only relevant for differentiating the side-to-side and up-and-down actuations, in which case it negates the thrust force in the vertical and horizontal directions, respectively. For the pulse and helix cases, the clocking angle only determines the phase of excitation and it is arbitrarily

set to 90° here. A pitching amplitude of  $A = 1.25^{\circ}$  is primarily used in this study, although other values of A are also discussed.

• Line 189 (and various other places): WES uses Eq. and Fig., not Equation and Figure as used by the authors throughout the paper.

Equation and Figure have been replaced with Eq. and Fig. throughout the manuscript.

- Line 199: Shouldn't it be  $\hat{u}_j(r, \kappa_{\theta}, \omega)$  instead of  $\theta$ ? Yes, thank you for noticing this error. It has been updated to  $\hat{u}_j(r, \kappa_{\theta}, \omega)$ .
- Figure 7: As mentioned before, this figure might not be necessary: A list/table of cross section locations would probably be clearer to me. Otherwise, the 3D representation does not seem to add much value to the figure and might even diminish clarity. Also, this paper uses normalized distance and centers around the turbine, whereas earlier figures used absolute distance. Please choose one or the other and apply throughout the paper (I would suggest using the one used here).

The streamwise location of the planes in Figure 7 of the original manuscript have been added to the figure caption so that it is explicitly clear which cross section locations are used in the SPOD analysis.

Line 212: period missing after "5".
 A period has been added.

• Figures 9 and 10: the panels in these figures are identical, but the subcaptions have very different sizes. Consider making these uniform.

Thank you for pointing out this typesetting error. The subcaptions are consistent in the updated version of these figures (see Figures 13 and 14 in this document)

 Figure 10, caption: typo, "w" instead of " " before "kappa"

This typo has been corrected in the manuscript.

- Line 269: I can see how the swirl due to blade rotation can induce some contribution to the -1 mode, but I would expect it mostly influences this mode at the 1P frequency, not the St=0.3 frequency. Have you investigated this?
  - $\rightarrow$  Line 270: Similarly, I can see how the veer contributes to the 1 mode, but would love to see a similar analysis in low-veer conditions to confirm this.
  - → Line 405: I would indeed be very interested to see how the CW helix performs here, as you suggest. I would highly recommend adding this case to the paper. If the main contribution of this paper was to find the best AWM strategy for power extraction, it makes complete sense to ignore this case. However, as this paper is more about understanding the aerodynamics of AWM, I think adding this case to the paper would strengthen the findings.

These three comments are excellent questions by the reviewer that the authors are hoping to address in a more thorough investigation in the future. Running additional LES cases is

not permitted for this project at the moment, and repeating the analysis in this manuscript for a low-veer environment is outside of its scope. However, the effects of veer and swirl on active wake mixing strategies have been documented in other recent studies [Brown et al., 2025, Frederik et al., 2025, Coquelet et al., 2024], and the authors hope to extend the modal analysis developed in this paper to provide an explanation for the behavior observed in these studies soon.

• Line 307: change "as shown below" to "as shown in Fig. 14), as the figure is actually on the following page.

This change has been made to the manuscript.

• Figure 13: I'm not sure how much value this figure adds. Panel b looks very similar to 11a, as one would expect. Panel a shows that your SPOD is working, but does not really match the overall point you are trying to make in this section. Furthermore, it makes me question: do we not have the same leading blade-rotation defined SPOD modes at 0.1D downstream? I would expect so, but you do not show that.

The reviewer is correct that Figure 13b looks similar to 11a in the original manuscript. This similarity arises because the blade pitch actuations induce a similar modal response in the induction field as in the near wake region at St = 0.3. However, unlike the near wake region, the dominant SPOD modes in the induction field do not occur at St = 0.3, but instead occur at the blade-passing frequency. This is not the case at 0.1D downstream, where the forced SPOD modes at St = 0.3 are dominant (Figure 11a in the original manuscript). The authors have decided to retain Figure 13 in the original manuscript to illustrate the difference (panel a) and similarity (panel b) between the modal structure of the induction field and the wake.

- Line 338: not "as", but "as opposed to"

  This has been corrected in the manuscript.
- Line 345: I'm not sure I agree that the runoff is steep enough that you only need the first 1-2 SPOD modes to accurately represent the flow. If you want to make this claim, you should plot the flow according to the first 2 SPOD modes and compare it to the actual flow.

The authors have revised the language regarding the eigenvalue decay in the near wake region for improved clarity. In the pulse and helix cases, an order-of-magnitude decrease in the eigenvalue spectrum is observed after the first three SPOD modes 0.5D downstream. In contrast, for the side-to-side and up-and-down cases, the decay is more gradual due to the forcing of two modes by the blade pitch actuations, resulting in an order-of-magnitude decrease after six SPOD modes.

The following text has been added to the manuscript:

To understand the separation of scales in the wake, the streamwise evolution of the eigenvalue spectra is reported in Fig. 14. In the pulse and helix cases, an order-of-magnitude decrease in the eigenvalue spectrum is observed after the first three SPOD modes at 0.5D downstream. In contrast, the side-to-side and up-and-down cases exhibit a more gradual decay, attributed to the forcing of two modes by the blade pitch actuations, resulting in an order-of-magnitude decrease after six SPOD modes. This suggests that the flow is effectively represented by the leading SPOD modes in the near wake region, where large-scale coherent structures dominate the flow's unsteadiness and account for the majority of the wake TKE. However, the decay

becomes more gradual starting around 3D downstream, and by 10D, the distribution of modal energy across all AWM cases is similar to the turbulence in the baseline wake.

- Line 352: "other wind farm control strategies"

  This change has been made to the manuscript.
- Figure 17: What does the tau in the subcaption of fig a mean?

On line 360 of the original manuscript where the figure is first referenced,  $\tau$  is defined as  $\tau = \frac{1}{2\pi} \int_0^{2\pi} \mathcal{T}(r=D/2,\theta) d\theta$ , i.e., it is a scalar measure of mean flow entrainment obtained by azimuthally averaging the radial shear stress flux around the circumference of a rotor disk.

This has been clarified in the caption of Figure 16:

Figure 16: (a) Circumferential average of the radial shear stress flux around a rotor disk centered at the wake center, defined as  $\tau = \frac{1}{2\pi} \int_0^{2\pi} \mathcal{T}(r = D/2, \theta) d\theta$ . (b) Rotor averaged velocity for each AWM case, normalized by the baseline value. The velocity is averaged over a rotor disk centered around the wake center.

• Figures 16 and 18: consider making the limits of these figures a little smaller, as these figures show a lot of grey area that provide no information now.

Thank you for this suggestion. The limits of Figures 16 and 18, as well as Figures 21 and 22, in the original manuscript have been adjusted, while maintaining the aspect ratio. For the reviewer's reference, the update version of Figure 16 of the original manuscript is included in Figure 16 of this document:

Figure 16: Contours of the radial shear stress flux for the baseline case at four different streamwise locations. The dashed line corresponds to the rotor disk centered at the hub height location and the dotted line corresponds to the rotor disk centered around the wake centers.

• Figure 20: Wouldn't it make more sense to plot —tau— here to make the figure smoother and more easily interpretable? Even if tau is negative, it still represents an eigenvalue, right?

The quantity  $\tau_{j,j}$  is not an eigenvalue; rather, it represents the contributions to the net radial shear stress flux from the jth SPOD mode. Specifically,  $\tau_{j,j}$  is positive for any mode that leads to a net turbulent entrainment of mean velocity into the wake, and negative for modes leading to the entrainment of mean velocity out of the wake. The scalar  $|\lambda_j|$  is the eigenvalue associated with the jth SPOD mode, and quantifies the turbulent kinetic energy associated with each mode. While it is possible to plot a monotonically decreasing  $\tau_{j,j}$  spectra it would require a re-shuffling of the SPOD mode indices j (compare the top and bottom rows of Figure 17 in this document). Not only would this be confusing, but it would eliminate any sort of comparison between the energy contents of a mode and the mode's contribution to

Figure 17: Energy and entrainment spectra defined by  $|\lambda_j|$  and  $\tau_{j,j}$ , respectively, for the leading 75 eigenvalues at  $(x - x_{hub})/D = 3$ .

wake recovery. The SPOD formulation only guarantees a monotonic ordering of modes based on eigenvalues, and there is no reason to expect  $\tau_{j,j}$  to line up with this ordering, especially for smaller turbulent scales. Throughout the manuscript, the modal index, j, is consistently ordered based on the SPOD eigenvalue, as is most natural, and  $\tau_{j,j}$  is plotted with respect to this ordering. The authors have clarified the definition of  $\tau_{j,j}$  in the manuscript and the intended purpose of Figure 20.

The following text has been included in the manuscript to clarify the quantity  $\tau_{jj}$  and the choice of SPOD parameters

- Similarly to  $\mathcal{T}$ , the net mean flow entrainment into the rotor disk from  $\mathcal{T}_{j,j}$  can be quantified by the scalar measure  $\tau_{j,j} = \frac{1}{2\pi} \int_0^{2\pi} \mathcal{T}_{j,j} (r = D/2, \theta) d\theta$ . While the eigenvalue,  $|\lambda_j|$ , is a scalar measure of modal TKE, the scalar quantity  $\tau_{j,j}$  represents the contributions to the net radial shear stress flux from the jth SPOD mode. The quantity  $\tau_{j,j}$  is positive for any mode that leads to a net turbulent entrainment of mean velocity into the wake, and negative for modes leading to the entrainment of mean velocity out of the wake.
- A comparison between modal energy,  $|\lambda_j|$ , and modal entrainment,  $\tau_{jj}$ , across a wider range of wavenumbers and frequencies is shown in Fig. 19. It is generally found that the dominant coherent structures in terms of TKE are also responsible for entraining the most mean flow. However,  $\tau_{j,j}$  does not strictly decrease with the eigenvalue index, j, like  $|\lambda_j|$  does; there are several instances across streamwise locations and AWM cases where the flow structure that entrains the most momentum is not the one with the largest modal energy, although it typically falls within the leading 5-10 eigenvalues. It is important to recall that the SPOD formulation only guarantees an ordering of modes based on energy, not entrainment, so there is no reason to expect, in general, that  $\tau_{j,j}$

decay monotonically with j like  $|\lambda_j|$ . This is particularly evident for the large j-indices shown in Fig. 19. Once the eigenvalue spectrum has decayed by roughly an order of magnitude, SPOD modes even begin to entrain mean flow out of the wake, as indicated by the negative values of  $\tau_{j,j}$ .

- Figure 19: Energy and entrainment spectra defined by  $|\lambda_j|$  and  $\tau_{j,j}$ , respectively, for the leading 75 eigenvalues at  $(x x_{hub})/D = 1$ , 3, and 6. The spectra are normalized on the maximum value of  $|\lambda_j|$  and  $\tau_{j,j}$  for each AWM case and streamwise location. Note that  $\tau_{j,j}$  is not a strictly positive quantity like  $|\lambda_j|$ , so rapid roll-offs in the spectra occur on a log-log scale for any modes contributing to a net entrainment of mean velocity out of the wake.
- Figures 21 and 22: if you want to keep these figures in the paper, you should include additional analysis of what you are showing here. It seems to me though that this analysis would not be in line with the main contribution of this paper, and is similar to previous LES studies performed on AWM. Furthermore, if you keep the figures, I would suggest using the same colormap throughout the paper (also applies to Fig 8).

The authors have opted to retain Figures 21 and 22 in the manuscript and add additional text surrounding these figures. One of the main contributions of the paper is connecting the SPOD of the actuated flow to the wake recovery, including the mean velocity deficit and the radial shear stress flux. The authors feel the contour figures are particularly important to show in addition to the integrated values to demonstrate the anisotropy of these quantities around the rotor disk, especially compared to the canonical case (e.g., [Cheung et al., 2024]).

The following section of manuscript has been updated:

The improvement in radial shear stress flux over the baseline is not homogeneous around the rotor disk, however, and there are points where mean flow is even lost compared to the baseline due to the AWM pitch actuation (see Fig. ??). This is a departure from the canonical case analyzed by Cheung et al. [2024], further highlighting the impact of ABL characteristics on wake mixing dynamics including wind veer, shear, and stratification. Moreover, the modal structure of the forcing is visible in the contours of  $\mathcal{T}$  within a turbine diameter downstream, where positive values are approximately axisymmetric for the pulse case but depend on the clocking angle for the up-and-down and side-to-side actuations.

Moreover, the authors have ensured that the colormaps in the manuscript are consistent. For any *diverging* quantity, such as the radial shear stress flux, a red-blue color scheme is used. For any *sequential* quantity, such as streamwise velocity, a viridis color scheme is used.

• Line 435: Please show some verification (or refer to a paper that shows) that this approach yields reliable results. I understand that it is not feasible to run simulations at each downstream distance and for each case, but you could at least use say two cases to show how well this method estimates the power capture of an LES with 2 turbines. You might also want to add a description of this approach in Section 2.1.

It is not feasible to run a two-turbine LES for this study; however, in lieu of this, it is common in the literature to estimate the power of a downstream turbine using the rotor averaged velocity (see, for example, [Taschner et al., 2024, Frederik et al., 2025]). As mentioned in an earlier response, we have chosen to approximate the power of the downstream turbine using an OpenFAST simulation coupled with inflow data from the LES from the appropriate

location, instead of just approximating power based on rotor-averaged velocity. This should be more accurate than the rotor-averaged velocity approach as it includes the turbine model's response to the LES inflow data, instead of just the power coefficient. The OpenFAST simulation also provides a measurement of the loads on the downstream turbine, which is an important consideration for wake control methods that cannot be evaluated from rotor-averaged velocity alone. The authors would characterize the approach as a good compromise between conducting a full LES and simply analyzing rotor-averaged velocities.

• Figure 23: consider adding in some way the results from Figure 17 in Figure 23 to make it easier for the reader to verify how well the rotor averaged velocity predicts downstream power. For this purpose, you might need to plot just the downstream power separately. Also, there seems to be a very thin grey line around this figure, is that done on purpose?

The authors have included the results for the downstream turbine's power in the updated version of this figure (see Figure 15 in this document). The rotor-averaged velocities are presented in a separate figure due to the slight vertical shift in the wake which the downstream turbine cannot follow; however, the trends are closely aligned. The authors appreciate the reviewer's suggestion and have also removed the mysterious thin grey line.

• Figure 24: If you do keep this figure, it makes no sense to me to unify the y-axes like you did here. The top 4 panels now provide close to no information as the differences are indistinguishably small. But similar to my main comment about Figure 23, I do not think this analysis adds much value to the paper and perhaps should be cut altogether.

Figure 24 of the original manuscript has been revised so that the y-axes are no longer unified across load channels (see Figure 18 in this document). Please note that some of the additional white space in the figures arises from the values of the DLC 1.2 case, which provides important context for the baseline loads.

Additionally, the following paragraph has been added to this section to clarify the DLC 1.2 results:

To provide context for these loads, the DELs for this turbine in a DLC 1.2 environment are also included in Fig. 18. A DLC 1.2 simulation is typically used to evaluate the structural integrity and performance of the wind turbine under expected operational conditions, ensuring that it can withstand the loads and stresses it will encounter during its operational life. The DLC 1.2 results are computed here with a single seed for the turbine using baseline controls in conditions derived from the Normal Turbulence Model with a hub height wind speed of 6.4 m/s and a turbulence intensity of 25.90%. This comparison is included to highlight that AWM is particularly well-suited for stable ABL conditions with low TI, where the significant increases in DELs over the baseline largely remain much smaller than the loads on the turbine in typical design environments.

**References**

Kenneth Brown, Gopal Yalla, Lawrence Cheung, Joeri Frederik, Nate deVelder, Dan Houck, Eric Simley, and Paul Fleming. Comparison of wind farm control strategies under a range of realistic wind conditions: wake quantities of interest. Wind Energy Science, 2025.

Figure 18: Baseline-normalized damage equivalent loads (DEL) for seven different load channels at three different pitching amplitudes,  $A = 0.5^{\circ}$ ,  $A = 1.25^{\circ}$ , and  $A = 2.0^{\circ}$ . Solid bars indicate DELs for the upstream turbine, while the DELs for the turbine 6D downstream are outlined in black. The red dashed line corresponds to the baseline-normalized DELs from a normal turbulence model in a DLC 1.2-like environment (single seed) with a hub height wind speed of 6.4 m/s, a shear exponent of 0.12, and a turbulence intensity of 25.90%.

Lawrence C Cheung, Kenneth A Brown, Daniel R Houck, and Nathaniel B deVelder. Fluid-dynamic mechanisms underlying wind turbine wake control with strouhal-timed actuation. Energies, 17 (4):865, 2024.

M Coquelet, J Gutknecht, JW Van Wingerden, M Duponcheel, and P Chatelain. Dynamic individual pitch control for wake mitigation: Why does the helix handedness in the wake matter? In Journal of Physics: Conference Series, volume 2767, page 092084. IOP Publishing, 2024.

Brandon L Ennis, Jonathan R White, and Joshua A Paquette. Wind turbine blade load characterization under yaw offset at the swift facility. In Journal of physics: Conference series, volume 1037, page 052001. IOP Publishing, 2018.

ARM Climate Research Facility. Southern great plains (sgp) observatory, 2024. URL https://www.arm.gov/capabilities/observatories/sgp. Accessed: 2022-02-02.

Joeri Frederik, Eric Simley, Kenneth Brown, Gopal Yalla, Lawrence Cheung, and Paul Fleming. Comparison of wind farm control strategies under a range of realistic wind conditions: turbine quantities of interest. Wind Energy Science, 2025.

Joeri A Frederik, Bart M Doekemeijer, Sebastiaan P Mulders, and Jan-Willem van Wingerden. The helix approach: Using dynamic individual pitch control to enhance wake mixing in wind farms. Wind Energy, 23(8):1739–1751, 2020.

Gregg Freebury and Walter Musial. Determining equivalent damage loading for full-scale wind turbine blade fatigue tests. In 2000 ASME wind energy symposium, page 50, 2000.

- Jonas Gutknecht, Marcus Becker, Claudia Muscari, Thorsten Lutz, and Jan-Willem van Wingerden. Scaling dmd modes for modeling dynamic induction control wakes in various wind speeds. In 2023 IEEE Conference on Control Technology and Applications (CCTA), pages 574–580. IEEE, 2023.
- Raghavendra Krishnamurthy, Rob K Newsom, Duli Chand, and William J Shaw. Boundary layer climatology at arm southern great plains. Technical report, Pacific Northwest National Lab.(PNNL), Richland, WA (United States), 2021.
- Patrick Moriarty, Nicola Bodini, Stefano Letizia, Aliza Abraham, Tyler Ashley, Konrad B Bärfuss, Rebecca J Barthelmie, Alan Brewer, Peter Brugger, Thomas Feuerle, et al. Overview of preparation for the american wake experiment (awaken). Journal of Renewable and Sustainable Energy, 16(5), 2024.
- C Muscari, P Schito, A Viré, A Zasso, D Van Der Hoek, and JW Van Wingerden. Physics informed dmd for periodic dynamic induction control of wind farms. In Journal of Physics: Conference Series, volume 2265, page 022057. IOP Publishing, 2022.
- National Renewable Energy Laboratory. Rosco v2.8.0, 2024. https://github.com/NREL/ROSCO.
- E Taschner, M Becker, Remco Verzijlbergh, and JW Van Wingerden. Comparison of helix and wake steering control for varying turbine spacing and wind direction. In Journal of Physics: Conference Series, volume 2767, page 032023. IOP Publishing, 2024.
- Aaron Towne, Oliver T Schmidt, and Tim Colonius. Spectral proper orthogonal decomposition and its relationship to dynamic mode decomposition and resolvent analysis. Journal of Fluid Mechanics, 847:821–867, 2018.
- Aemilius Adrianus Wilhelmus van Vondelen, Marion Coquelet, Sachin Tejwant Navalkar, and Jan-Willem van Wingerden. Synchronized helix wake mixing control. Wind Energy Science Discussions, 2025:1–36, 2025.

---

## Author Response (AR2)

We thank both reviewers for their second review of the manuscript and for recommending the paper for publication. We are pleased that most of the initial comments from the reviewers were addressed in the revised manuscript. Two further edits have been made to address the remaining comments from both reviewers, which are detailed below.

1. First, we agree with the second reviewer that the analysis of the baseline case at different Strouhal numbers is a worthy result for publication, and it has been included in the revised manuscript. In particular, Figure 1 of this document is now discussed at the beginning of Section 3, and then the remainder of the results focus in on St = 0.3, as this corresponds to the dominant frequency in the baseline wake and the selected AWC forcing frequency.

Figure 1: Eigenvalues of **S** at five different Strouhal numbers for the baseline case. Shown are the azimuthal wavenumbers  $\kappa_{\theta} = 0$  (—),  $\kappa_{\theta} = 1$  (—),  $\kappa_{\theta} = -1$  (—),  $\kappa_{\theta} = 2$  (—), and  $\kappa_{\theta} = -2$  (—). Each eigenvalue is normalized by the global maximum baseline eigenvalue across all streamwise locations and Strouhal numbers.

The utility of SPOD for identifying the optimal AWC forcing frequency a priori based on baseline wake data is also suggested, and the authors feel that a more detailed investigation of wind turbine wakes within the range  $St \in [0.25, 0.35]$  across different atmospheric conditions would be worthwhile for future research. We note that the SPOD parameters selected in this study do not permit for such fine resolution.

Please see the latest version of the manuscript for the revised text, highlighted in blue.

2. Second, it seems the analysis of the two-turbine array presented in Section 3.3 has caused some confusion and we apologize for the miscommunication. Upon revisiting the paper after a few months, we agree with the second reviewer that the analysis of power and loads for different pitching amplitudes does not align well with the primary focus of the manuscript, which centers on  $A = 1.25^{\circ}$ . Additionally, the results at  $A = 0.5^{\circ}$  and  $2^{\circ}$  may introduce questions that detract from the overall narrative of the paper.

However, we do believe there is a value in connecting the spectral analysis of turbine loads, wake TKE, and turbulent entrainment that is discussed in the preceding sections to practical quantities of interest, specifically the power gains of a downstream turbine and conventional loading metrics for the upstream turbine.

Therefore, we considerably streamlined the discussion in Section 3.3. Figures 22 and 23 have been combined into a single figure (see Figure 2 in this document), which focuses exclusively

Figure 2: (top) Percent change in generated power, P, from the baseline case  $(P_{A=0^{\circ}})$  for a two turbine array with the turbine spacings ranging from 1D to 14D. The results for the upstream turbine are computed from the LES, while the downstream turbine results are obtained from a stand-alone OpenFAST simulation using YZ-planes from the LES as inflow data at each streamwise location. AWM is applied to the upstream turbine with  $A=1.25^{\circ}$ , while the downstream turbine is operated using baseline controls. (bottom) Baseline-normalized damage equivalent loads (DEL) for seven different load channels. Solid bars indicate DELs for the upstream turbine, while the DELs for the turbine 5D downstream are outlined in black. The red dashed line corresponds to the DELs of the turbine using baseline controls in DLC 1.2 conditions (single seed) derived from the Normal Turbulence Model with a hub height wind speed of 6.4 m/s (i.e., turbulence intensity of 25.90%).

on the two-turbine array results for the  $A=1.25^{\circ}$  cases. In this figure, the change in power from the baseline case is shown for the two-turbine array, as well as for the downstream and upstream turbines individually. Moreover, the DELs for seven different load channels are also shown for the upstream turbine and a downstream turbine. Note, in the revised manuscript, DELs are shown for a turbine spaced 5D downstream, as this corresponds to the location where the combined power increase is maximized. In addition to the revised figure, the discussion in this section has been significantly shortened to focus on the main cases of interest.

Even though the  $A=0.5^{\circ}$  and  $A=2^{\circ}$  results are no longer included in the manuscript, we would still like to respond to the reviewers comments about these cases to the best of our ability. The three main comments about this section are discussed below:

• Power gains for the helix at  $A = 0.5^{\circ}$ : We do not have a great explanation yet for

the small increase in power observed for the helix case at small pitching amplitudes. The first reviewer may be correct that this is due to a lack of statistical convergence for the  $A=0.5^{\circ}$  LES cases. However, we would like to note that this trend is something that we have occasionally observed in other simulations when using small pitch amplitudes, even with different turbine models and wind conditions. Nonetheless, more work is needed to understand if this result is significant or not, and so we have decided to omit these findings from the current manuscript.

- Loss in power for the up-and-down case loses more power than the other strategies at  $A=2^{\circ}$ , even producing less power than the baseline for the two turbine array. We believe this is a result of the airfoil operating in a suboptimal region of the lift curve when a large pitching amplitude is applied to the turbine blade in the vertical  $0^{\circ}$  position. The vertical position is where the angle-of-attack is typically near its maximum, especially in the presence of a highly sheared inflow. A similar discussion can be found in the work of Taschner et al. [2023].
- CHANGES IN REPORTED VALUES: The reviewers are correct that a few numerical values in Section 3.3 have changed from the original version of the manuscript. In the initial version, a slightly different averaging window was applied to all the AWC cases and the baseline case. Specifically, given a time series of data on the interval  $t \in [t_0, t_1]$ , the following averaging windows were initially used for all cases:

```
Baseline: t \in [t_0, t_1]

Pulse: t \in [t_0, t_0 + \lfloor (t_1 - t_0)/(2\pi/\omega_e) \rfloor \times 2\pi/\omega_e]

Helix: t \in [t_0, t_0 + \lfloor (t_1 - t_0)/(2\pi/(\bar{\Omega} - \omega_e)) \rfloor \times 2\pi/(\bar{\Omega} - \omega_e)]

Side-to-side: t \in [t_0, t_0 + \lfloor (t_1 - t_0)/(\pi/\omega_e) \rfloor \times \pi/(\omega_e)]

Up-and-down: t \in [t_0, t_0 + \lfloor (t_1 - t_0)/(\pi/\omega_e) \rfloor \times \pi/(\omega_e)]
```

where  $\omega_e = 2\pi StU_{\infty}/D$  is the excitation frequency, and  $\bar{\Omega}$  is the mean rotor speed. This approach was taken to closely match the period of oscillation or the "beat-envelope" of each AWC case, as described by Cheung et al. [2024]. However, this difference in the averaging window led to inconsistencies when comparing AWC cases and in the reported baseline-normalized values, especially in the presence of a non-uniform inflow, which was not an issue in [Cheung et al., 2024]. Moreover, one of the numerical values was not updated correctly in the revised text. In the current manuscript, we have ensured that the power and DEL results reported in Section 3.3 are averaged over seven equal Strouhal periods after the transients from the stand-alone OpenFAST simulations are discarded, and this information is included in the text. Averaging over an equal number of Strouhal periods accounts for the large fluctuations in power introduced by the pulse method [Frederik et al., 2020] while also ensuring consistency in the time intervals across cases.

Please see the latest version of the manuscript for the revised text, highlighted in blue, and note that much of the text in Section 3.3 has been removed from earlier versions.

**References**

- Lawrence C Cheung, Kenneth A Brown, Daniel R Houck, and Nathaniel B deVelder. Fluid-dynamic mechanisms underlying wind turbine wake control with strouhal-timed actuation. Energies, 17 (4):865, 2024.
- Joeri A Frederik, Bart M Doekemeijer, Sebastiaan P Mulders, and Jan-Willem van Wingerden. The helix approach: Using dynamic individual pitch control to enhance wake mixing in wind farms. Wind Energy, 23(8):1739–1751, 2020.
- Emanuel Taschner, Aemilius AW van Vondelen, Remco Verzijlbergh, and Jan-Willem van Wingerden. On the performance of the helix wind farm control approach in the conventionally neutral atmospheric boundary layer. In Journal of Physics: Conference Series, volume 2505, page 012006. IOP Publishing, 2023.